# Optimal Decision Tree with Noisy Outcomes

**Su Jia**[*]
Carnegie Mellon University
sjia1@andrew.cmu.edu

**Fatemeh Navidi**[†]
University of Michigan
navidi@umich.edu

**Viswanath Nagarajan**
University of Michigan
viswa@umich.edu

**R. Ravi**
Carnegie Mellon University
ravi@andrew.cmu.edu

## Abstract

A fundamental task in active learning involves performing a sequence of tests to identify an unknown hypothesis that is drawn from a known distribution. This problem, known as optimal decision tree induction, has been widely studied for decades and the asymptotically best-possible approximation algorithm has been devised for it. We study a generalization where certain test outcomes are noisy, even in the more general case when the noise is persistent, i.e., repeating a test gives the same noisy output, disallowing simple repetition as a way to gain confidence. We design new approximation algorithms for both the non-adaptive setting, where the test sequence must be fixed *a-priori*, and the adaptive setting where the test sequence depends on the outcomes of prior tests. Previous work in the area assumed at most a logarithmic number of noisy outcomes per hypothesis and provided approximation ratios that depended on parameters such as the minimum probability of a hypothesis. Our new approximation algorithms provide guarantees that are nearly best-possible and work for the general case of a large number of noisy outcomes per test or per hypothesis where the performance degrades smoothly with this number. Our results adapt and generalize methods used for submodular ranking and stochastic set cover. We evaluate the performance of our algorithms on two natural applications with noise: toxic chemical identification and active learning of linear classifiers. Despite our theoretical logarithmic approximation guarantees, our methods give solutions with cost very close to the information theoretic minimum, demonstrating the effectiveness of our methods.

## 1  Introduction

The classic optimal decision tree (ODT) problem involves identifying an initially unknown hypothesis $\bar{x}$ that is drawn from a known probability distribution over a set of $m$ possible hypotheses. We can perform tests in order to distinguish between these hypotheses. Each test produces a binary outcome (positive or negative) and the precise outcome of each hypothesis-test pair is known beforehand. [3] So an instance of ODT can be viewed as a $\pm 1$ matrix with the hypotheses as rows and tests as columns. The goal is to identify hypothesis $\bar{x}$ using the minimum number of tests in expectation.

As a motivating application, consider the following task in medical diagnosis [25]. A doctor needs to diagnose a patient's disease by performing tests. Given an *a priori* probability distribution over possible diseases, what sequence of tests should the doctor perform? Another application is in active

---

[*]Su Jia and Fatemeh Navidi contributed equally to this work.

[†]Research of F. Navidi and V. Nagarajan partly supported by NSF grant CCF-1750127.

[3]We consider binary test outcomes only for simplicity: our results also hold for finitely many outcomes.

learning [10]. Given a set of data points, one wants to learn a classifier that labels the points correctly as positive and negative. There is a set of $m$ possible classifiers which is assumed to contain the true classifier. In the Bayesian setting, which we consider, the true classifier is drawn from some known probability distribution. The goal is to identify the true classifier by querying labels at the minimum number of points (in expectation). Other applications include entity identification in databases [6] and experimental design to choose the most accurate theory among competing candidates [14].

An important issue that is not considered in the classic ODT model is that of unknown or noisy outcomes. In fact, our research was motivated by a dataset involving toxic chemical identification where the outcomes of many hypothesis-test pairs are stated as unknown (one of our experimental results is also on this dataset). Prior work incorporating noise in ODT [14] is restricted to settings with very sparse noise. In this paper, we design approximation algorithms for the noisy optimal decision tree problem in full generality.

We consider a standard model for persistent noise. Certain outcomes (i.e., entries in the hypothesis-test matrix) are random with a known distribution: for simplicity we treat each noisy outcome as an unbiased $\pm 1$ random variable. Our results extend directly to the case when each noisy outcome has a different probability of being $\pm 1$. Persistent noise means that repeating the same test always produces the same $\pm 1$ outcome. We assume that the instance is identifiable, i.e., a unique hypothesis can always be identified irrespective of the noisy outcomes. (This assumption can be relaxed: see §6.)

We consider both non-adaptive policies (where the test sequence is fixed upfront) and adaptive policies (where the test sequence is built incrementally and depends on observed test outcomes). Clearly, adaptive policies perform at least as well as non-adaptive ones.[4] However, non-adaptive policies are very simple to implement (requiring minimal incremental computation) and may be preferred in time-sensitive applications. Our main contributions are:

- an $O(\log m)$-approximation algorithm for non-adaptive ODT with noise.
- an $O(\min(h, r) + \log m)$-approximation algorithm for adaptive ODT with noise, where $h$ (resp. $r$) is the maximum number of noisy outcomes for any hypothesis (resp. test).
- an $O(\log m)$-approximation algorithm for adaptive ODT with noise when every test has at least $m - O(\sqrt{m})$ noisy outcomes.
- experimental results on applications to toxic chemical identification and active learning.

We note that both non-adaptive and adaptive versions (even for usual ODT) generalize the set cover problem: so an $\Omega(\log m)$ approximation ratio is the best possible (unless P=NP).

**Related Work**  The optimal decision tree problem (without noise) has been extensively studied for several decades [11, 20, 25, 24, 1, 2, 7, 18]. The state-of-the-art result [18] is an $O(\log m)$-approximation, for instances with arbitrary probability distribution and costs. It is also known that ODT cannot be approximated to a factor better than $O(\log m)$, unless P=NP [6].

The application of ODT to Bayesian active learning was formalized in [10]. There are also several results on the *statistical complexity* of active learning. e.g. [4, 19, 29], where the focus is on proving bounds for structured hypothesis classes. In contrast, we consider arbitrary hypothesis classes and obtain *computationally efficient* policies with provable approximation bounds relative to the optimal (instance specific) policy. This approach is similar to that in [10, 15, 12, 14, 9, 22].

The noisy ODT problem was studied previously in [14]. Using a connection to adaptive-submodularity [12], they obtained an $O(\log^2 \frac{1}{p_{min}})$-approximation algorithm for noisy ODT in the presence of very few noisy outcomes; here $p_{min} \leq \frac{1}{m}$ is the minimum probability of any hypothesis.[5] In particular, the running time of the algorithm in [14] is exponential in the number of noisy outcomes per hypothesis, which is polynomial only if this number is at most logarithmic in the number of hypotheses/tests. Our result provides the following improvements (i) the running time is polynomial irrespective of the number of noisy outcomes and (ii) the approximation ratio is better by at least one logarithmic factor. We note that a better $O(\log m)$ approximation ratio (still only for very sparse noise) follows from subsequent work on the "equivalence class determination" problem by [9].

For this setting, our result is also an $O(\log m)$ approximation, but the algorithm is simpler. More importantly, ours is the first result that can handle *any* number of noisy outcomes.

Other variants of noisy ODT have also been considered, e.g. [27, 5, 8], where the goal is to identify the correct hypothesis with at least some target probability. The theoretical results in [8] provide "bicriteria" approximation bounds where the algorithm has a larger error probability than the optimal policy. Our setting is different because we require *zero* probability of error.

Many algorithms for ODT (including ours) rely on some underlying submodularity properties. We briefly survey some background results. The basic submodular cover problem was first considered by [31], who proved that the natural greedy algorithm is a $(1 + \ln \frac{1}{\epsilon})$-approximation algorithm, where $\epsilon$ is the minimal positive marginal increment of the function. [3] obtained an $O(\log \frac{1}{\epsilon})$-approximation algorithm for the submodular ranking problem, that involves simultaneously covering multiple submodular functions; [21] extended this result to also handle costs. [23] studied an adaptive version of the submodular ranking problem. We utilize results/techniques from these papers.

Finally, we note that there is also work on minimizing the *worst-case* (instead of average case) cost in ODT and active learning [26, 30, 16, 17]. These results are incomparable to ours because we are interested in the average case, i.e. minimizing expected cost.

## 2 Problem Definition

We start with defining the optimal decision tree with noise (ODTN) formally. There is a set of $m$ possible hypotheses with a probability distribution $\{\pi_x\}_{x=1}^m$, from which an unknown hypothesis $\bar{x}$ is drawn. There is also a set $U = [n]$ of binary tests. Each test $e \in U$ is associated with a 3-way partition $T^+(e), T^-(e), T^*(e)$ of the hypotheses, where the test outcome is (a) positive if $\bar{x}$ lies in $T^+(e)$, (b) negative if $\bar{x} \in T^-(e)$, and (c) positive or negative with probability $\frac{1}{2}$ each if $\bar{x} \in T^*(e)$ (these are noisy outcomes). We assume that conditioned on $\bar{x}$, each noisy outcome is independent. We also use $r_x(e)$ to denote the part of test $e$ that hypothesis $x$ lies in, i.e.

$$r_x(e) = \begin{cases} -1 & \text{if } x \in T^-(e) \\ +1 & \text{if } x \in T^+(e) \\ * & \text{if } x \in T^*(e) \end{cases}$$

While we know the 3-way partition $T^+(e), T^-(e), T^*(e)$ for each test $e \in U$ upfront, we are not aware of the actual outcomes for the noisy hypothesis-test pairs. It is assumed that the realized hypothesis $\bar{x}$ can be uniquely identified by performing all tests, regardless of the outcomes of $*$-tests. This means that for every pair $x, y \in [m]$ of hypotheses, there is some test $e \in U$ with $x \in T^+(e)$ and $y \in T^-(e)$ or vice-versa. The goal is to perform an adaptive (or non-adaptive) sequence of tests to identify hypothesis $\bar{x}$ using the minimum *expected* number of tests. Note that expectation is taken over both the prior distribution of $\bar{x}$ and the random outcomes of noisy tests for $\bar{x}$.

In our algorithms and analysis, it will be convenient to work with an **expanded set** of hypotheses $M$. For a binary vector $b \in \{\pm 1\}^U$ and hypothesis $x \in [m]$, we say $b$ is consistent with $x$ and denote $b \sim x$, if $b_e = r_x(e)$ for each $e \in U$ with $r_x(e) \neq *$. Let $M = \{(b, x) \in \{\pm 1\}^U \times [m] : b \sim x\}$, and $M_x \subseteq M$ be all copies associated with a particular hypothesis $x \in [m]$; note that $\{M_x\}_{x=1}^m$ is a partition of $M$. Each "expanded" hypothesis $(b, x) \in M$ corresponds to the case where the true hypothesis $\bar{x} = x$ and the test-outcomes are given by $b$. We assign the probability $q_{b,x} = \pi_x / 2^{h_x}$ to each $(b, x) \in M$, where $h_x$ is the number of *-tests for $x$. Note that conditioned on $\bar{x} = x$, the probability of observing outcomes $b$ is exactly $2^{-h_x}$; so $\Pr[\bar{x} = x$ and test outcomes are $b] = q_{b,x}$. For any $(b, x) \in M$ and $e \in U$, define $r_{b,x}(e) = b(e)$ to be the observed outcome of test $e$ if $\bar{x} = x$ and test-outcomes are $b$. For every expanded hypothesis $(b, x) \in M$ and test $e \in U$, define

$$T_{b,x}(e) = \begin{cases} T^+(e) & \text{if } r_{b,x}(e) = -1 \\ T^-(e) & \text{if } r_{b,x}(e) = +1 \end{cases}, \tag{1}$$

which is the subset of (original) hypotheses that can *definitely* be ruled-out based on test $e$ if $\bar{x} = x$ and the test-outcomes are given by $b$. Note that hypotheses in $T^*(e)$ are never part of $T_{b,x}(e)$ as their outcome on test $e$ can be positive/negative (so they cannot be ruled-out). For every hypothesis $(b, x) \in M$, define a monotone submodular function $f_{b,x} : 2^U \to [0, 1]$:

$$f_{b,x}(S) = |\bigcup_{e \in S} T_{b,x}(e)| \cdot \frac{1}{m-1}, \qquad \forall S \subseteq U, \tag{2}$$

which equals the fraction of the $m - 1$ hypotheses (excluding $x$) that have been ruled-out based on the tests in $S$ if $\bar{x} = x$ and test-outcomes are given by $b$. Assuming $\bar{x} = x$ and test-outcomes are given by $b$, hypothesis $x$ is uniquely identified after tests $S$ if and only if $f_{b,x}(S) = 1$.

A **non-adaptive** policy is specified by just a permutation of tests. The policy performs tests in this sequence and eliminates incompatible hypotheses until there is a unique compatible hypothesis (which is $\bar{x}$). Note that the number of tests performed under such a policy is still random (depends on $\bar{x}$ and outcomes of noisy tests). An **adaptive** policy chooses tests incrementally, depending on prior test outcomes. The *state* of a policy is a tuple $(E, d)$ where $E \subseteq U$ is a subset of tests and $d \in \{\pm 1\}^E$ denotes the observed outcomes on tests in $E$. An adaptive policy is specified by a mapping $\Phi : 2^U \times \{\pm 1\}^U \to U$ from states to tests, where $\Phi(E, d)$ is the next test to perform at state $(E, d)$. Equivalently, we can view a policy as decision tree with nodes corresponding to states, labels at nodes representing the test performed at that state and branches corresponding to the $\pm 1$ outcome at the current test. As the number of states can be exponential, we cannot hope to specify arbitrary adaptive policies. Instead, we want implicit policies $\Phi$, where given *any* state $(E, d)$, the test $\Phi(E, d)$ can be computed *efficiently*. This would imply that the total time taken under any outcome is polynomial. We note that an optimal policy $\Phi^*$ can be very complex and the map $\Phi^*(E, d)$ may not be efficiently computable. We will still compare the performance of our (efficient) policy to $\Phi^*$.

In this paper, we consider the **persistent noise** model. That is, repeating a test $e$ with $\bar{x} \in T^*(e)$ always produces the same outcome. An alternative model is non-persistent noise, where each run of test $e$ with $\bar{x} \in T^*(e)$ produces an independent random outcome. The persistent noise model is more appropriate to handle missing data. It also contains the non-persistent noise model as a special case (by introducing multiple tests with identical partitions). One can easily obtain an adaptive $O(\log^2 m)$-approximation for the non-persistent model using existing algorithms for noiseless ODT [7] and repeating each test $O(\log m)$ times. The persistent-noise model that we consider is much harder.

## 3 Non-Adaptive Algorithm

Our algorithm is based on a reduction to the submodular ranking problem [3], defined below.

**Submodular Function Ranking (SFR)** An instance of SFR consists of a ground set $U$ of elements and a collection of monotone submodular functions $\{f_1, ..., f_m\}$, $f_x : 2^U \to [0, 1]$, with $f_x(\emptyset) = 0$ and $f_x(U) = 1$ for all $x \in [m]$. Additionally, there is a weight $w_x \geq 0$ for each $x \in [m]$. A solution is a permutation of the elements $U$. Given any permutation $\sigma$ of $U$, the **cover time** of function $f$ is $C(f, \sigma) := \min\{t \,|\, f(\cup_{i \in [t]} \sigma(i)) = 1\}$ where $\sigma(i)$ is the $i^{th}$ element in $\sigma$. In words, it is the earliest time when the value of $f$ reaches the unit threshold. The goal is to find a permutation $\sigma$ of $[n]$ with minimal total cover time $\sum_{x \in [m]} w(x) \cdot C(f_x, \sigma)$. We will use the following result:

**Theorem 3.1** ([3]). *There is an $O(\log \frac{1}{\epsilon})$-approximation for SFR where $\epsilon$ is minimum marginal increment of any function.*

The non-adaptive ODTN problem can be expressed as an instance of SFR as follows. The elements are the tests $U$. For each hypothesis-copy $(b, x) \in M$ there is a function $f_{b,x}$ (see (2)) with weight $q_{b,x}$. Based on the definition of these functions, the parameter $\epsilon = \frac{1}{m-1}$. To see the equivalence, note that a solution to non-adaptive ODTN is also a permutation $\sigma$ of $U$ and hypothesis $x$ is uniquely identified under outcome $(b, x)$ exactly when function $f_{b,x}$ has value one. Moreover, the objective of the ODTN problem is the expected number of tests in $\sigma$ to identify the realized hypothesis $\bar{x}$, which equals

$$\sum_{x=1}^{m} \pi_x \sum_{b \sim x} 2^{-h_x} \cdot C_{b,x}(\sigma) = \sum_{(b,x) \in M} q_{b,x} \cdot C_{b,x}(\sigma),$$

where $C_{b,x}(\sigma)$ is the cover-time of function $f_{b,x}$. It now follows that this SFR instance is equivalent to the non-adaptive ODTN instance. However, we cannot apply Theorem 3.1 directly to obtain an $O(\log m)$ approximation. This is because we have an exponential number of functions (note $|M|$ can be exponential in $m$), which means that a direct implementation of the algorithm from [3] requires exponential time. Nevertheless, we show that a variant of the SFR algorithm can be used to obtain:

**Theorem 3.2.** *There is an $O(\log m)$-approximation for non-adaptive ODTN.*

The SFR algorithm [3] is a greedy-style algorithm that at any point, having already chosen tests $E$, assigns a score to each test $e \in U \setminus E$ of

$$G_E(e) := \sum_{(b,x) \in M: f_{b,x}(E) < 1} q_{b,x} \frac{f_{b,x}(\{e\} \cup E) - f_{b,x}(E)}{1 - f_{b,x}(E)} = \sum_{(b,x) \in M} q_{b,x} \cdot \Delta_E(b, x, e), \quad (3)$$

$$\Delta_E(b, x, e) = \begin{cases} \frac{f_{b,x}(\{e\} \cup E) - f_{b,x}(E)}{1 - f_{b,x}(E)}, & \text{if } f_{b,x}(E) < 1; \\ 0, & \text{otherwise.} \end{cases} \quad (4)$$

where $\Delta_E(b, x, e)$ is the "gain" of test $e$ for the hypothesis-copy $b, x$. At each step, the algorithm chooses the test of maximum score. However, we do not know how to compute the score (3) in polynomial time. Instead, using the fact that $G_E(e)$ is the expectation of $\Delta_E(b, x, e)$ over the hypothesis-copies $(b, x) \in M$, we will show that we can obtain an approximate maximizer by sampling. Moreover, Theorem 3.1 also holds when we choose a test with approximately maximum score: this follows directly from the analysis in [21]. This sampling approach is still not sufficient because it can fail when the value $G_E(e)$ is very small. A key observation is, when the score $G_E(e)$ is small for all tests $e$ then it must be that, with high probability the already-performed tests $E$ uniquely identify hypothesis $\bar{x}$. Hence the future tests won't affect the expected cover time by much.

As the realized hypothesis $\bar{x}$ can always be identified uniquely, for any pair $x, y \in [m]$ of hypotheses, there is a test where $x$ and $y$ have opposite outcomes (i.e. one is $+$ and the other $-$). So there is a set $\mathcal{L}$ of at most $\binom{m}{2}$ tests where hypothesis $\bar{x}$ will be uniquely identified by performing all the tests in $\mathcal{L}$.

The non-adaptive ODTN algorithm (Non-Adap) involves two phases. In the first phase, we run the SFR algorithm using sampling to get estimates $\bar{G}_E(e)$ of the scores $G_E(e)$ at each step; let $e^* = \arg\max_{e \in U} \bar{G}_E(e)$ denote the chosen test. If at some step, the maximum sampled score is less than $m^{-5}$ then we go to the second phase where we perform all the tests in $\mathcal{L}$ and stop. The number of samples used to obtain each estimate is polynomial in $m$; so the overall runtime is polynomial. The complete proof can be found in the full version of this paper.

## 4 Adaptive Algorithms

Our adaptive algorithm chooses between two algorithms ($ODTN_r$ and $ODTN_h$) based on the noise sparsity parameters $h$ (maximum number of noisy outcome per hypothesis), and $r$ (maximum number of noisy outcome per test). These two algorithms maintain the posterior probability of each hypothesis based on the previous test outcomes, and use these probabilities to calculate a "score" for each test. The score of a test has two components (i) a term that prioritizes splitting the candidate hypotheses in a balanced way and (ii) terms that correspond to the expected number of hypotheses eliminated. We maintain the following information at each point in the algorithm: already performed tests $E \subseteq U$, compatible hypotheses $H \subseteq [m]$ and (posterior) probability $p_x$ for each $x \in H$. Given values $\{p_x : x \in [m]\}$, to reduce notation we use the shorthand $p(S) = \sum_{x \in S} p_x$ for any subset $S \subseteq [m]$. The main difference between the two algorithms we have, is in the defining the metric for component (i). First we discuss $ODTN_r$:

---
**Algorithm 1** ODTN$_r$
---
initially $E \leftarrow \emptyset$, $H \leftarrow [m]$ and $p_x \leftarrow \pi_x$ for all $x \in [m]$.
**while** $|H| > 1$ **do**
    for any test $e \in U$, let $L_e(H)$ be the smaller cardinality set among $T^+(e) \cap H$ and $T^-(e) \cap H$
    select test $e \in U \setminus E$ that maximizes:

$$p(L_e(H)) + \frac{|T^-(e) \cap H|}{|H| - 1} \cdot p(T^+(e) \cap H) + \frac{|T^+(e) \cap H|}{|H| - 1} \cdot p(T^-(e) \cap H) + \frac{|H \setminus T^*(e)|}{2(|H| - 1)} \cdot p(T^*(e) \cap H). \quad (5)$$

    if outcome of test $e$ is $+$ then $H \leftarrow H \setminus T^-(e)$; else $H \leftarrow H \setminus T^+(e)$.
    $E \leftarrow E \cup \{e\}$ and update $p_x \leftarrow p_x/2$ for all $x \in T^*(e)$.
**end while**
---

**Theorem 4.1.** *Algorithm 1 is an $\mathcal{O}(r + \log m)$-approximation algorithm for adaptive ODTN, where $r$ is the maximum number of noisy outcomes per test.*

The high-level idea is to view any ODT instance $\mathcal{I}$ as a suitable instance $\mathcal{J}$ of adaptive submodular ranking (ASR). Then we will use and modify an existing framework of analysis of ASR from [23].

**An equivalent ASR instance $\mathcal{J}$.** This involves the expanded hypothesis set $M$ where each hypothesis $(b, x) \in M$ occurs with probability $q_{b,x} = \pi_x / 2^{h_x}$. Each hypothesis $(b, x)$ is also associated with: (i) submodular function $f_{b,x} : 2^U \to [0, 1]$ and (ii) feedback function $r_{b,x} : U \to \{+, -\}$ where $r_{b,x}(e)$ is the outcome of test $e$ under hypothesis $(b, x)$. The goal in the ASR instance is to adaptively select a subset $S \subseteq U$ such that the value $f_{b,x}(S) = 1$ for the realized hypothesis $(b, x)$. The objective is to minimize the expected cost $\mathbb{E}[|S|]$.

**Lemma 4.2.** *The ASR instance $\mathcal{J}$ is equivalent to ODT instance $\mathcal{I}$.*

Now, we present an algorithm for the ASR instance $\mathcal{J}$ that we will show is equivalent to running Algorithm 1 on instance $\mathcal{I}$. Crucially, the ASR algorithm is almost identical to that studied in prior work [23] and therefore we can essentially re-use the analysis from that paper to prove our bound. Recall that the expanded hypotheses $M = \cup_{x=1}^m M_x$ where $M_x$ are all copies of hypothesis $x \in [m]$. To reduce notation, we use $q(S) = \sum_{(b,x) \in S} q_{b,x}$ for any subset $S \subseteq M$. Also note that hypothesis $(b, x)$ is covered when $f_{b,x}(E) = 1$ which implies identifying hypothesis $x \in [m]$.

---

**Algorithm 2** Algorithm for ASR instance $\mathcal{J}$.

initially $E \leftarrow \emptyset, H' \leftarrow M$.
**while** $H' \neq \emptyset$ **do**
    $H \leftarrow \{x \in [m] : M_x \cap H' \neq \emptyset\}$.
    for any test $e \in U$, let $L'_e(H') = \{(b, x) \in H' : x \in L_e(H)\} = H' \cap \left( \cup_{x \in L_e(H)} M_x \right)$
    select test $e \in U \setminus E$ that maximizes:

$$q\left(L'_e(H')\right) \; + \sum_{(b,x) \in M_x \cap H'} q_{b,x} \cdot \frac{f_{b,x}(e \cup E) - f_{b,x}(E)}{1 - f_{b,x}(E)}. \tag{6}$$

    remove incompatible and covered hypotheses from $H'$ based on the feedback from $e$.
    $E \leftarrow E \cup \{e\}$
**end while**

---

We now prove the equivalence between Algorithms 1 and 2. The *state* of either algorithm is represented by the set $E$ of tests performed along with their outcomes.

**Lemma 4.3.** *The decision tree produced by Algorithm 1 on $\mathcal{I}$ is the same as that produced by Algorithm 2 on $\mathcal{J}$.*

Based on Lemmas 4.2 and 4.3, in order to prove Theorem 4.1 it suffices to show that Algorithm 2 is an $O(r + \log m)$-approximation algorithm for ASR instance $\mathcal{J}$. The proof is very similar to the analysis in [23]. So we only provide an outline of the overall proof, while emphasizing the differences. For $k = 0, 1, \cdots$, define the following quantities:

- $A_k \subseteq M$ is the set of uncovered hypotheses in ALG at time $L \cdot 2^k$, and $a_k = q(A_k)$.
- $Y_k$ is the set of uncovered hypotheses in OPT at time $2^{k-1}$, and $y_k = q(Y_k)$.

Here $L = O(r + \log m)$. The key step is to show:

$$a_k \leq 0.2 a_{k-1} + 3 y_k, \qquad \text{for all } k \geq 1. \tag{7}$$

As shown in [23], this implies an $O(L)$ approximation ratio. In order to prove (7) we use the quantity:

$$Z := \sum_{t > L2^{k-1}}^{L2^k} \sum_{(E,H') \in R(t)} \max_{e \in U \setminus E} \left( \sum_{(b,x) \in L'_e(H')} q_{b,x} \; + \sum_{(b,x) \in H'} q_{b,x} \cdot \frac{f_{b,x}(e \cup E) - f_{b,x}(E)}{1 - f_{b,x}(E)} \right) \tag{8}$$

Above, $R(t)$ denotes the set of states $(E, H')$ that occur at time $t$ in ALG. (7) will be proved by separately lower and upper bounding $Z$.

**Lemma 4.4** ([23]). *We have $Z \geq L \cdot (a_k - 3y_k)/3$.*

**Lemma 4.5.** *We have $Z \leq a_{k-1} \cdot (1 + \ln m + r + \log m)$.*

*Proof.* For any hypothesis $(b, x) \in A_{k-1}$ (i.e. uncovered in ALG by time $L2^{k-1}$) let $\sigma_{b,x}$ be the path traced by $(b, x)$ in ALG's decision tree, starting from time $2^{k-1}L$ and ending at $2^k L$ or when $(b, x)$ gets covered. Recall that for any $L2^{k-1} < t \leq L2^k$, any hypothesis in $H'$ for any state in $R(t)$ appears in $A_{k-1}$. So only hypotheses in $A_{k-1}$ can contribute to $Z$ and we rewrite (8) as:

$$
\begin{aligned}
Z \quad &= \sum_{(b,x) \in A_{k-1}} q_{b,x} \cdot \sum_{e \in \sigma_{b,x}} \left( \frac{f_{b,x}(e \cup E) - f_{b,x}(E)}{1 - f_{b,x}(E)} + \mathbb{1}[(b,x) \in L'_e(H')] \right) \\
&\leq \sum_{(b,x) \in A_{k-1}} q_{b,x} \cdot \left( \sum_{e \in \sigma_{b,x}} \frac{f_{b,x}(e \cup E) - f_{b,x}(E)}{1 - f_{b,x}(E)} + \sum_{e \in \sigma_{b,x}} \mathbb{1}[(b,x) \in L'_e(H')] \right) \quad (9)
\end{aligned}
$$

Above, for any $e \in \sigma_{b,x}$ we use $(E, H')$ to denote the state at which $e$ is selected.

Fix any hypothesis $(b, x) \in A_{k-1}$. For the first term, we use Lemma 4.6 below and the definition of $\epsilon$. This implies $\sum_{e \in \sigma_{b,x}} \frac{f_{b,x}(e \cup E) - f_{b,x}(E)}{1 - f_{b,x}(E)} \leq 1 + \ln \frac{1}{\epsilon} \leq 1 + \ln m$ as parameter $\epsilon \geq 1/m$ for $f_{b,x}$.

Now, we bound the second term by proving the inequality below:

$$
\sum_{e \in \sigma_{b,x}} \mathbb{1}[(b,x) \in L'_e(H')] \leq r + \log m \quad (10)
$$

To prove this inequality, consider hypotheses in $H'$. Now, if hypothesis $(b, x) \in L'_e(H')$ when ALG selects test $e$, then $x$ would be in $L_e(H)$. Suppose $L_e(H) = T^+(e) \cap H$; the other case is identical. Let $D_e(H) = T^-(e) \cap H$ and $S_e(H) = T^*(e) \cap H$. As $x \in L_e(H)$, it must be that path $\sigma_{b,x}$ follows the + branch out of $e$. Also, the number of candidate hypotheses on this path after test $e$ is

$$
|L_e(H)| + |S_e(H)| \leq \frac{|L_e(H)|}{2} + \frac{|D_e(H)|}{2} + |S_e(H)| = \frac{|H|}{2} + \frac{|S_e(H)|}{2} \leq \frac{|H|}{2} + \frac{r}{2}.
$$

The first inequality uses the definition of $L_e(H)$ and the last inequality uses the bound of $r$ on the number of hypotheses with $*$ outcomes. Hence, each time that $(b, x) \in L'_e(H')$ along path $\sigma_{b,x}$, the number of candidate hypotheses changes as $|H_{new}| \leq \frac{1}{2}|H_{old}| + \frac{r}{2}$. This implies that after $\log_2 m$ such events, $|H| \leq r$. Let $\sigma'_{b,x}$ denote the portion of path $\sigma_{b,x}$ after $|H|$ drops below $r$. Note that each time $(b, x) \in L'_e(H')$ we have $L_e(H) \neq \emptyset$: so $|H|$ reduces by at least one after each such test $e$. Hence $\sum_{e \in \sigma'_{b,x}} \mathbb{1}[(b,x) \in L'_e(H')] \leq r$. As the portion of path $\sigma_{b,x}$ until $|H| \leq r$ contributes at most $\log_2 m$ to the left-hand-side in (10), the total is at most $r + \log_2 m$ as needed. $\square$

**Lemma 4.6** ([3]). *Let $f : 2^U \to [0, 1]$ be any monotone function with $f(\emptyset) = 0$ and $\epsilon = \min\{f(S \cup \{e\}) - f(S) : e \in U, S \subseteq U, f(S \cup \{e\}) - f(S) > 0\}$. Then, for any sequence $\emptyset = S_0 \subseteq S_1 \subseteq \cdots S_k \subseteq U$ of subsets, we have $\sum_{t=1}^{k} \frac{f(S_t) - f(S_{t-1})}{1 - f(S_{t-1})} \leq 1 + \ln \frac{1}{\epsilon}$.*

Setting $L = 15(1 + \ln m + r + \log_2 m)$ and applying Lemmas 4.4 and 4.5 completes the proof of (7) and hence Theorem 4.1.

We now discuss algorithm $ODTN_h$. This is based on directly applying the ASR algorithm from [23] to $\mathcal{J}$. The resulting algorithm is very similar to Algorithm 2 and involves a change in the definition of $L'_e(H')$ in score (6) to be the smaller of the following sets:

$$
\{(b,x) \in H' : r_{b,x}(e) = +\} \quad \text{and} \quad \{(b,x) \in H' : r_{b,x}(e) = -\}.
$$

The main difference in the analysis is in proving the following inequality instead of inequality (10) in the proof of Lemma 4.5:

$$
\sum_{e \in \sigma_{b,x}} \mathbb{1}[(b,x) \in L'_e(H')] \leq h + \log m \quad (11)
$$

This follows from the observation that each time that $(b, x) \in L'_e(H')$ along path $\sigma_{b,x}$, the size $|H'|$ reduces by at least a factor $\frac{1}{2}$ (note that initially $|H'| = |M| \leq 2^h \cdot m$). Finally by choosing the best between $ODTN_r$ and $ODTN_h$, we obtain:

**Theorem 4.7.** *There is an $\mathcal{O}(\min(h,r) + \log m)$-approximation algorithm for adaptive ODTN, where $h$ (resp. $r$) is the maximum number of noisy outcomes per hypothesis (resp. test).*

We note that our analysis is tight (up to constant factors).

In order to understand the dependence of the approximation ratio on the noise sparsity $\min(h,r)$, we study instances that have a very large number of noisy outcomes. Formally, we define an $\alpha$-**sparse** ($\alpha \leq 1/2$) instance as follows. There is a constant $C$ such that $\max\{|T^+(e)|, |T^-(e)|\} \leq C \cdot m^\alpha$ for all tests $e \in U$. Somewhat surprisingly, there is a very different algorithm (see the full version) that can actually take advantage of the large noise:

**Theorem 4.8.** *There is an adaptive $O(\log m)$-approximation for ODTN on $\alpha \leq \frac{1}{2}$ sparse instances.*

## 5 Non-identifiable Instances

We have assumed that for every pair $x, y$ of hypotheses, there is some test that distinguishes them deterministically. Without this assumption, we can still obtain similar results by slightly changing the stopping criterion. Define a *similarity graph* $G$ on $m$ nodes (corresponding to hypotheses) with an edge $(x, y)$ if there is *no* test separating $x$ and $y$ deterministically. Let $D_x$ denote the set containing $x$ and all its neighbors in $G$, for each $x \in [m]$. The *neighborhood* stopping criterion involves stopping when the set $H$ of compatible hypotheses is contained in *some* $D_x$, where $x$ might or might not be $\bar{x}$. The *clique* stopping criterion involves stopping when $H$ is contained in some clique of $G$. Note that clique stopping is a stronger notion of identification than neighborhood stopping. Our algorithms' performance guarantees will now also depend on the maximum degree $d$ of $G$; note that $d = 0$ in the perfectly identifiable case. We obtain a non-adaptive algorithm with approximation ratio $O((d+1)\log m)$ and an adaptive algorithm with approximation ratio $O(d + \min(h,r) + \log m)$. Below we outline our approach for the adaptive algorithm; the details can be found in the full version.

For the adaptive version, we run a two-phase algorithm. In the first phase, we identify some subset $N \subseteq [m]$ containing $\bar{x}$ with $|N| \leq d$. This can be done using the algorithm in §4 with the following submodular function for each $(b, x) \in M$.

$$f_{b,x}(S) = |\bigcup_{e \in S} T_{b,x}(e)| \cdot \frac{1}{m - d}, \qquad \forall S \subseteq U.$$

The expected cost of this phase is $O(\min(r,h) + \log m) \cdot OPT$ using an analysis identical to §4; here $OPT$ denotes the optimal value. Then, in the second phase, we run a simple splitting algorithm that iteratively selects any test that splits the current set $H$ of candidate scenarios, until the neighborhood or clique stopping criterion is satisfied. The expected cost of this phase is at most $d \cdot OPT$. Combining both phases, we obtain an $O(d + \min(h,r) + \log m)$-approximation algorithm.

## 6 Extensions

**Non-binary outcomes.** We can also handle tests with an arbitrary set $\Sigma$ of outcomes (instead of $\pm 1$). This requires extending the outcomes $b$ to be in $\Sigma^U$ and applying this change to the definitions of sets $T_{b,x}$ (1) and submodular function $f_{b,x}$ (2).

**Non-uniform noise distribution.** Our results extend directly to the case where each noisy outcome has a different probability of being $\pm 1$. Suppose that the probability of every noisy outcome is between $\delta$ and $1 - \delta$. Then Theorems 3.2 and 4.7 continue to hold (irrespective of $\delta$), and Theorem 4.8 holds with a slightly worse $O(\frac{1}{\delta} \log m)$ approximation ratio.

## 7 Experiments

We implemented our algorithms, and performed experiments on real-world and synthetic data sets. We compared our algorithms' cost (expected number of tests) with an information theoretic lower bound on the optimal cost and show that the difference is negligible. Thus, despite our logarithmic approximation ratios, the practical performance can be much better.

**Chemicals with Unknown Test Outcomes**    One natural application of ODT is identifying chemical or biological materials. We considered a data set called WISER[6], which includes 400+ chemicals (hypothesis) and 78 binary tests. Every chemical has either positive, negative or unknown result on each test. We have performed our algorithms on both the original instance, in which some chemicals are not perfectly identifiable, and a modified version. In the modified version, to ensure every pair of chemicals can be distinguished, we removed the chemicals that are not identifiable from each other to be left with 255 chemicals (to do this, we used a greedy rule that iteratively drops the highest-degree hypothesis in the similarity graph).

**Random Binary Classifiers with Margin Error**    We construct a dataset containing 100 two-dimensional points, by picking each of their attributes uniformly in $[-1000, 1000]$. We also choose 2000 random triples $(a, b, c)$ to form linear classifiers $\frac{ax+by}{\sqrt{a^2+b^2}} + c \leq 0$, where $a, b \leftarrow N(0, 1)$ and $c \leftarrow U(-1000, 1000)$. The point labels are binary and we introduce noisy outcomes based on the distance of each point to a classifier. Specifically, for each threshold $d \in \{0, 5, 10, 20, 30\}$ we define dataset CL-$d$ that has a noisy outcome for any classifier-point pair where the distance of the point to the boundary of the classifier is smaller than $d$. In order to ensure that the instances are perfectly identifiable, we remove "equivalent" classifiers and we are left with 234 classifiers.

**Algorithms**    We implement the following algorithms: the adaptive $O(r + \log m)$-approximation (ODTN$_r$), the adaptive $O(h + \log m)$-approximation (ODTN$_h$), the non-adaptive $O(\log m)$-approximation (Non-Adap) and a slightly adaptive version of Non-Adap (Low-Adap). Algorithm Low-Adap considers the same sequence of tests as Non-Adap while (adaptively) skipping non-informative tests based on observed outcomes. The implementations of the adaptive and non-adaptive algorithms are available online.[7] We also consider three different stopping criteria: *unique* stopping for perfectly identifiable instances, *neighborhood* and *clique* stopping (defined in Section 5) for original WISER dataset.

**Results**    Table 1 shows the results of different algorithms with unique stopping on all identifiable datasets when the distribution over hypothesis is uniform, and Table 2 summarizes the results on original WISER with other stopping criteria. Experiments with some non-uniform distributions are presented in the supplementary material. Table 1 also reports values of an information theoretic lower bound (the entropy $\log_2 m$) on the optimal cost (Low-BND). We can see that ODTN$_r$ consistently outperforms the other algorithms and is very close to the lower bound. Note that original WISER dataset that is used to produce results in Table 2 has $m = 414$ hypotheses and $d = 54$ in its similarity graph, while the processed WISER in Table 1 is perfectly identifiable with $m = 255$ hypotheses.

| Data(r,h) / Algorithm | WISER (245,45) | CL-0 (0,0) | CL-5 (5,3) | CL-10 (7,6) | CL-20 (12,8) | CL-30 (13,8) |
|---|---|---|---|---|---|---|
| **Low-BND** | **7.994** | **7.870** | **7.870** | **7.870** | **7.870** | **7.870** |
| ODTN$_r$ | 8.357 | 7.910 | 7.927 | 7.915 | 7.962 | 8.000 |
| ODTN$_h$ | 9.707 | 7.910 | 7.979 | 8.211 | 8.671 | 8.729 |
| Non-Adap | 11.568 | 9.731 | 9.831 | 9.941 | 9.996 | 10.204 |
| Low-Adap | 9.152 | 8.619 | 8.517 | 8.777 | 8.692 | 8.803 |

Table 1: Cost of Algorithms with Unique Stopping for Uniform Distribution. For each dataset we also indicate the noise parameters $h$ and $r$ (max number of noisy outcomes per hypothesis/test).

| Algorithm | Neighborhood Stopping | Clique Stopping |
|---|---|---|
| ODTN$_r$ | 11.163 | 11.817 |
| ODTN$_h$ | 11.908 | 12.506 |
| Non-Adap | 16.995 | 21.281 |
| Low-Adap | 16.983 | 20.559 |

Table 2: Cost of Algorithms on original WISER dataset with Neighborhood and Clique Stopping for Uniform Distribution.

## Footnotes

[4] There are also instances where the relative gap between the best adaptive and non-adaptive policies is $\tilde{\Omega}(m)$.

[5] The paper [14] states the approximation ratio as $O(\log 1/p_{min})$ because it relied on an erroneous claim in [12]. The correct approximation ratio, based on [28, 13], is $O(\log^2 1/p_{min})$.

[6]https://wiser.nlm.nih.gov

[7]https://github.com/FatemehNavidi/ODTN ; https://github.com/sjia1/ODT-with-noisy-outcomes

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
