[Supplementary Material · long_version.pdf]

# Optimal Decision Tree with Noisy Outcomes

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

We use the following sampling lemma (Chernoff bound).

**Lemma 3.3.** *Let $X$ be a $[0,1]$ bounded random variable with $\mathbb{E}X \geq \Omega(m^{-5})$. Let $\bar{X}$ denote the average of $m^6$ many independent samples of $X$. Then $\Pr\left[\bar{X} \notin [\frac{1}{2}\mathbb{E}X, 2\mathbb{E}X]\right] \leq e^{-\Omega(m)}$.*

*Proof.* Let $X_1, ..., X_N$ be i.i.d. samples of random variable $X$ where $N = m^6$ is the number of samples. Letting $Y = \sum_{i \in [N]} X_i$, the usual Chernoff bound implies for any $\delta \in (0, 1)$,

$$\Pr\left(Y \notin [(1-\delta)\mathbb{E}Y, (1+\delta)\mathbb{E}Y]\right) \leq \exp(-\frac{\delta^2}{2} \cdot \mathbb{E}Y).$$

Setting $\delta = \frac{1}{2}$ and using the assumption $\mathbb{E}Y = N \cdot \mathbb{E}X = \Omega(m)$, the lemma follows. $\square$

**Lemma 3.4.** *Consider any step with $\bar{S} := \max_{e \in U} \bar{G}_E(e)$. If $\bar{S} \geq m^{-5}$ then $G_E(e^*) \geq \frac{S}{2}$.*

*Proof.* Let $S := \max_{e \in U} G_E(e)$. We first claim that $S \geq \frac{1}{2}m^{-5}$ w.h.p: otherwise, Lemma 3.3 would imply $\bar{S} < m^{-5}$ (we can artificially increase the mean to satisfy the "large mean" assumption in that lemma, which only makes proving the upper tail harder).

Consider first any $e \in U$ with $G_E(e) < S/4$. By Lemma 3.3, it follows that $\Pr[\bar{G}_E(e) > S/2] \leq e^{-\Omega(m)}$. Now consider the test $e'$ with $G_E(e') = S$. Again, by Lemma 3.3, it follows that $\Pr[\bar{G}_E(e') \leq S/2] \leq e^{-\Omega(m)}$. This means that w.h.p., test $e^*$ has $\bar{G}_E(e^*) \geq \bar{G}_E(e') > S/2$ and again by Lemma 3.3, $G_E(e^*) > S/4$. $\square$

**Lemma 3.5.** *Consider the step in our algorithm with $\max_{e \in U} \bar{G}_E(e) < m^{-5}$. Then the probability that the realized hypothesis $\bar{x}$ is uniquely identified by the tests $E$ is $1 - m^{-3}$.*

*Proof.* We prove the contrapositive. Suppose that the probability $z$ of *not* identifying the realized hypothesis $\bar{x}$ after $E$ is more than $m^{-3}$. Let $p_x(y) = \Pr_{b \sim x}[y \text{ not ruled out by } E | \bar{x} = x]$ denote the probability that when $\bar{x} = x$, hypothesis $y$ is not ruled out after performing tests $E$. Note that

$$z = \sum_{x=1}^m \pi_x \cdot \Pr_{b \sim x}[E \text{ doesn't rule out } [m] \setminus x] \leq \sum_{x=1}^m \pi_x \sum_{y \in [m] \setminus x} p_x(y) \leq m \sum_{x=1}^m \left(\pi_x \cdot \max_{y \in [m] \setminus x} p_x(y)\right)$$

It follows that there is some $x \in [m]$ with $\pi_x \cdot \max_{y \in [m] \setminus x} p_x(y) \geq \frac{z}{m^2}$. So there is some $x, y \in [m]$ with $\pi_x \cdot p_x(y) \geq \frac{z}{m^2} \geq m^{-5}$, where we used the assumption that $z \geq m^{-3}$. Recall that there is some test $e^*$ that separates $x$ and $y$ deterministically. Let $B' = \{b \sim x : y \notin \cup T_{b,x}(e)\}$. Note that

$\sum_{b \in B'} q_{b,x} = \pi_x \cdot p_x(y)$. For any $b \in B'$ we have (i) $y \notin \cup T_{b,x}(e)$ $f_{b,x}(E)$ and (ii) $y \in T_{b,x}(e^*)$ (this is true for *all* $b \sim x$). Therefore $\Delta_E(b, x, e^*) \geq \frac{1}{m-1}$ for all $b \in B'$, which implies:

$$g_E(e^*) \geq \sum_{b \sim x} q_{b,x} \Delta_E(b, x, e^*) \geq \sum_{b \in B'} q_{b,x} \frac{1}{m-1} = \frac{\pi_x \cdot p_x(y)}{m-1} \geq m^{-5},$$

as desired. $\square$

**Proof of Theorem 3.2.** We bound the expected costs (number of tests) from phase 1 and 2 separately. By Lemma 3.4, the test chosen in each step of phase 1 is a 2-approximate maximizer (w.h.p.) of the score used in the ASR algorithm. So the expected cost in phase 1 is at most $O(\log m)$ times the optimum. By Lemma 3.5, the probability of running phase 2 is at most $m^{-3}$. As there are $|\mathcal{L}| \leq m^2$ tests in phase 2, the expected cost is $o(1)$. So we obtain an $O(\log m)$-approximation algorithm.

# 4 Adaptive Algorithms

Our adaptive algorithm maintains the posterior probability of each hypothesis based on the previous test outcomes, and uses these probabilities to calculate a "score" for each test. The score of a test has two components (i) a term that prioritizes splitting the candidate hypotheses in a balanced way and (ii) terms that correspond to the expected number of hypotheses eliminated. We maintain the following information at each point in the algorithm: already performed tests $E \subseteq U$, compatible hypotheses $H \subseteq [m]$ and (posterior) probability $p_x$ for each $x \in H$. Given values $\{p_x : x \in [m]\}$, to reduce notation we use the shorthand $p(S) = \sum_{x \in S} p_x$ for any subset $S \subseteq [m]$.

---

**Algorithm 1** ODTN$_r$

---
    initially $E \leftarrow \emptyset$, $H \leftarrow [m]$ and $p_x \leftarrow \pi_x$ for all $x \in [m]$.
    **while** $|H| > 1$ **do**
        for any test $e \in U$, let $L_e(H)$ be the smaller cardinality set among $T^+(e) \cap H$ and $T^-(e) \cap H$
        select test $e \in U \setminus E$ that maximizes:

$$p(L_e(H)) + \frac{|T^-(e) \cap H|}{|H| - 1} \cdot p\left(T^+(e) \cap H\right) + \frac{|T^+(e) \cap H|}{|H| - 1} \cdot p\left(T^-(e) \cap H\right) + \frac{1}{2} \frac{|H \setminus T^*(e)|}{|H| - 1} \cdot p\left(T^*(e) \cap H\right). \tag{5}$$

        **if** outcome of test $e$ is $+$ **then**
            update $H \leftarrow H \setminus T^-(e)$
        **else**
            update $H \leftarrow H \setminus T^+(e)$
        **end if**
        $E \leftarrow E \cup \{e\}$
        **for** $x \in H$ **do**
            **if** $r_x(e) = *$ **then**
                $p_x \leftarrow p_x/2$
            **end if**
        **end for**
    **end while**

---

**Theorem 4.1.** *Algorithm 1 is an $\mathcal{O}(r + \log m)$-approximation algorithm for adaptive ODTN, where $r$ is the maximum number of noisy outcomes per test.*

The high-level idea is to view any ODT instance $\mathcal{I}$ as a suitable instance $\mathcal{J}$ of adaptive submodular ranking (ASR). Then we will use an existing framework of analysis of ASR from [24].

**An equivalent ASR instance $\mathcal{J}$.** This involves the expanded hypothesis set $M$ where each hypothesis $(b, x) \in M$ occurs with probability $q_{b,x} = \pi_x / 2^{h_x}$. Each hypothesis $(b, x)$ is also associated with: (i) submodular function $f_{b,x} : 2^U \to [0, 1]$ and (ii) feedback function $r_{b,x} : U \to \{+, -\}$ where $r_{b,x}(e)$ is the outcome of test $e$ under hypothesis $(b, x)$. The goal in the ASR instance is to adaptively select a subset $S \subseteq U$ such that the value $f_{b,x}(S) = 1$ for the realized hypothesis $(b, x)$. The objective is to minimize the expected cost $\mathbb{E}[|S|]$.

**Lemma 4.2.** *The ASR instance $\mathcal{J}$ is equivalent to ODT instance $\mathcal{I}$.*

*Proof.* We will show that any feasible decision tree for $\mathcal{J}$ (resp. $\mathcal{I}$) is also feasible for instance $\mathcal{I}$ (resp. $\mathcal{I}$) with the same objective. In one direction, let $\mathcal{T}$ be a decision tree for $\mathcal{J}$. For any hypothesis $(b, x) \in M$ let $P_{b,x}$ denote the unique path traced in $\mathcal{T}$ and let $S_{b,x}$ denote the tests performed. Then we have $f_{b,x}(S_{b,x}) = 1$ which means $\bigcup_{e \in S_{b,x}} T_{b,x}(e) = [m] \setminus x$. Now consider $\mathcal{T}$ as a decision tree for the ODT instance $\mathcal{I}$. *Condition* on hypothesis $x \in [m]$ and outcomes $b$ on the $*$-tests for $x$, which occurs with probability $q_{b,x} = \pi_x / 2^{h_x}$. Then, it is clear that the feedback from any test $e$ is $r_{b,x}(e)$ and so the path traced in $\mathcal{T}$ is just $P_{b,x}$. Moreover, the set of incompatible hypotheses based on test $e$ is $T_{b,x}(e)$. So the set of incompatible hypotheses at the end of $P_{b,x}$ is $\bigcup_{e \in S_{b,x}} T_{b,x}(e) = [m] \setminus x$, which means $x$ is identified. Taking an expectation over all $x$ and $b$, it follows that $\mathcal{T}$ is a feasible decision tree for $\mathcal{I}$ with cost at most that for instance $\mathcal{J}$.

In the other direction, let $\mathcal{T}'$ be any decision tree for instance $\mathcal{I}$. Again *condition* on hypothesis $x \in [m]$ and outcomes $b$ on the $*$-tests for $x$ (with probability $q_{b,x}$). Then a unique path $P'_{b,x}$ is traced in $\mathcal{T}'$, and let $S'_{b,x}$ denote the tests on this path. As before, the set of incompatible hypotheses at the end of $P'_{b,x}$ is $\bigcup_{e \in S'_{b,x}} T_{b,x}(e) = [m] \setminus x$ because $x$ is identified. Now consider $\mathcal{T}'$ as a decition tree for ASR instance $\mathcal{J}$. Under hypothesis $b, x$, it is clear that path $P'_{b,x}$ is traced and so tests $S'_{b,x}$ are selected. It follows that $f_{b,x}(S'_{b,x}) = 1$ which means that hypothesis $b, x$ is covered at the end of $P'_{b,x}$. So $\mathcal{T}'$ is a feasible decision tree for $\mathcal{J}$. Taking expectations, the cost for $\mathcal{J}$ is at most that for instance $\mathcal{I}$. $\qquad\square$

Now, we present an algorithm for the ASR instance $\mathcal{J}$ that we will show is equivalent to running Algorithm 1 on instance $\mathcal{I}$. Crucially, the ASR algorithm is almost identical to that studied in prior work [24] and therefore we can essentially re-use the analysis from that paper to prove our bound. Recall that the expanded hypotheses $M = \cup_{x=1}^m M_x$ where $M_x$ are all copies of hypothesis $x \in [m]$. To reduce notation, we use $q(S) = \sum_{(b,x) \in S} q_{b,x}$ for any subset $S \subseteq M$. Also note that hypothesis $(b, x)$ is covered when $f_{b,x}(E) = 1$ which implies identifying hypothesis $x \in [m]$.

---

**Algorithm 2** Algorithm for ASR instance $\mathcal{J}$.

    initially $E \leftarrow \emptyset, H' \leftarrow M$.
    **while** $H' \neq \emptyset$ **do**
        $H \leftarrow \{x \in [m] : M_x \cap H' \neq \emptyset\}$.
        for any test $e \in U$, let $L'_e(H') = \{(b, x) \in H' : x \in L_e(H)\} = H' \cap \left( \cup_{x \in L_e(H)} M_x \right)$
        select test $e \in U \setminus E$ that maximizes:

$$q\left(L'_e(H')\right) \;+\; \sum_{(b,x) \in M_x \cap H'} q_{b,x} \cdot \frac{f_{b,x}(e \cup E) - f_{b,x}(E)}{1 - f_{b,x}(E)}. \tag{6}$$

        remove incompatible and covered hypotheses from $H'$ based on the feedback from $e$.
        $E \leftarrow E \cup \{e\}$
    **end while**

---

We now prove the equivalence between Algorithms 1 and 2. The *state* of either algorithm is represented by the set $E$ of tests performed along with their outcomes.

**Lemma 4.3.** *If Algorithms 1 and 2 are at the same state (i.e. performed the same set of tests $E$ and observed the same outcomes) then the set $H \subseteq [m]$ in the two algorithms is identical.*

*Proof.* Let $H_1$ and $H_2$ denote the set $H$ in Algorithms 1 and 2 respectively. We will show that $H_1 = H_2$. Consider any $x \in H_1$. We must have observed $r_x(e)$ on every test $\{e \in E : r_x(e) \neq *\}$. As there is a scenario-copy corresponding to *every* outcome on the $*$-tests for $x$, we will have some $(b, x)$ that is compatible with the observations on all tests in $E$. This means $(b, x) \in H'$ and so $x \in H_2$. On the other hand, consider $x \in H_2$, i.e. there is some $(b, x) \in H'$. Then we must have observed $r_{b,x}(e)$ on every test $e \in E$. In particular, we must have observed $r_x(e)$ on each $\{e \in E : r_x(e) \neq *\}$, which means that $x \in H_1$. $\qquad\square$

**Lemma 4.4.** *If Algorithms 1 and 2 are at the same state (i.e. performed the same set of tests $E$ and observed the same outcomes) then $p_x = \sum_{(b,x) \in H' \cap M_x} q_{b,x}$ for all $x \in H$.*

*Proof.* This can be proved by induction. Note that at the beginning we clearly have $p_x = \pi_x = \sum_{(b,x) \in M_x} q_{b,x}$ for all $x \in [m]$. Consider any state in the two algorithms when we have performed tests $E$ and observed the same outcomes. Let $H \subseteq [m]$ denote the compatible hypotheses at this state (which is the same in both algorithms by Lemma 4.3) and $e$ be the next test performed under both algorithms. Let $\bar{H} \subseteq H$ and $\bar{H}' \subseteq H'$ denote the compatible hypotheses after this test. We will prove inductively the expression for $p_x$ for any $x \in \bar{H}$ by considering different cases for $r_x(e)$.

If $r_x(e) \neq *$ and the outcome of $e$ is different from $r_x(e)$ then $x \notin \bar{H}$ and there is nothing to prove.

If $r_x(e) \neq *$ and the outcome of $e$ is $r_x(e)$ then $x \in \bar{H}$ and $r_{b,x}(e) = r_x(e)$ for all copies $(b,x)$ of $x$. So $M_x \cap H' = M_x \cap \bar{H}'$ and the expression for $p_x$ remains the same.

If $r_x(e) = *$ then $p_x$ decreases by a factor of 2 in Algorithm 1. By definition of the copies $M_x$ of hypothesis $x$, it is clear that exactly half the copies $(b,x) \in M_x \cap H'$ have $r_{b,x}(e) = +$ and the rest have $r_{b,x}(e) = -$. So, irrespective of the observed outcome of $e$, we have $|M_x \cap \bar{H}'| = \frac{1}{2}|M_x \cap H|$. This implies $q\left(\bar{H}' \cap M_x\right) = \frac{\pi_x}{2^{h_x}}|\bar{H}' \cap M_x| = \frac{1}{2}q\left(\bar{H} \cap M_x\right) = p_x$ (after the update). $\square$

**Lemma 4.5.** *Consider any state $(E, H')$ of Algorithm 2 and $(b,x) \in H'$. The following are true:*

1. $\bigcup_{d \in E} T_{b,x}(d) = [m] \setminus H$. So $f_{b,x}(E) = \frac{m-|H|}{m-1}$.

2. *For any $e \in U$, $f_{b,x}(E \cup e) - f_{b,x}(E) = \frac{|H \cap T_{b,x}(e)|}{m-1}$.*

3. *If $x \in H \cap T^+(e)$ then $f_{b,x}(E \cup e) - f_{b,x}(E) = \frac{|H \cap T^-(e)|}{m-1}$.*

4. *If $x \in H \cap T^-(e)$ then $f_{b,x}(E \cup e) - f_{b,x}(E) = \frac{|H \cap T^+(e)|}{m-1}$.*

5. *If $x \in H \cap T^*(e)$ then*

$$\sum_{(b,x) \in H'_x} q_{b,x}\left(f_{b,x}(E \cup e) - f_{b,x}(E)\right) = \frac{1}{2}\sum_{(b,x) \in H'_x} q_{b,x}\left(\frac{|H \cap T^-(e)|}{m-1} + \frac{|H \cap T^+(e)|}{m-1}\right),$$

*where $H'_x = M_x \cap H'$.*

*Proof.* (1) As $(b,x) \in H'$, the outcome of each test $d \in E$ must have been $r_{b,x}(d)$. By definition, $T_{b,x}(d)$ consists of all hypotheses $y \in [m]$ with $r_y(d) \neq *$ and $r_y(d) \neq r_{b,x}(d)$. So $\bigcup_{d \in E} T_{b,x}(d) \subseteq [m]$ is precisely the set of incompatible hypotheses at this state. In other words, $\bigcup_{f \in E} T_{b,x}(f) = [m] \setminus H$ and $f_{b,x}(E) = \frac{|\bigcup_{d \in E} T_{b,x}(d)|}{m-1} = \frac{m-|H|}{m-1}$.

(2) The set of hypotheses in $H$ that are compatible with $(b,x)$ after test $e$ are $H \setminus T_{b,x}(e)$. So based on (1) we can write:

$$f_{b,x}(E \cup e) - f_{b,x}(E) = \frac{m - |H \setminus T_{b,x}(e)|}{m-1} - \frac{m - |H|}{m-1} = \frac{|H \setminus T_{b,x}(e)| - |H|}{m-1} = \frac{|H \cap T_{b,x}(e)|}{m-1}.$$

(3-4) These follow directly from (2) using the definition of $T_{b,x}(e)$.

(5) It is easy to see (as observed before) that half the scenario-copies $b, x \in H' \cap M_x = H'_x$ have $r_{b,x}(e) = +$ (which implies $T_{b,x}(e) = T^-(e)$) and the rest have $r_{b,x}(e) = -$ (which implies $T_{b,x}(e) = T^+(e)$). So using (2),

$$\sum_{(b,x) \in H'_x} q_{b,x}\left(f_{b,x}(E \cup e) - f_{b,x}(E)\right) = \sum_{\substack{(b,x) \in H'_x \\ b(e)=-}} q_{b,x}\left(\frac{|H \cap T^+(e)|}{m-1}\right) + \sum_{\substack{(b,x) \in H'_x \\ b(e)=+}} q_{b,x}\left(\frac{|H \cap T^-(e)|}{m-1}\right),$$

which equals the claimed expression as exactly half the copies in $H'_x$ have $b(e) = +$. $\square$

**Lemma 4.6.** *The decision tree produced by Algorithm 1 on $\mathcal{I}$ is the same as that produced by Algorithm 2 on $\mathcal{J}$.*

*Proof.* Consider any state which is common to both algorithms, with $E$ being the already chosen tests, $H \subseteq [m]$ the compatible hypotheses and $H' \subseteq M$ the compatible uncovered scenario-copies. We will prove:

1. The score of each test $e$ is the same for both algorithms (at this state). Therefore the same test is chosen in both algorithms.

2. The stopping criteria for both algorithms is the same, i.e. Algorithm 1 stops at this state iff Algorithm 2 stops.

This clearly suffices to prove the lemma.

*Proving 1:* We need to show that (5) and (6) are equal. Consider any $x \in H$. By Lemma 4.4, we have $p_x = \sum_{(b,x) \in H'_x} q_{b,x}$. Now using Lemma 4.5, we have the following three cases:

- If $x \in H \cap T^+(e)$ then

$$\sum_{(b,x) \in H'_x} q_{b,x} \left( \frac{f_{b,x}(E \cup e) - f_{b,x}(E)}{1 - f_{b,x}(E)} \right) = p_x \cdot \frac{|H \cap T^-(e)|}{|H| - 1}.$$

- If $x \in H \cap T^-(e)$ then

$$\sum_{(b,x) \in H'_x} q_{b,x} \left( \frac{f_{b,x}(E \cup e) - f_{b,x}(E)}{1 - f_{b,x}(E)} \right) = p_x \cdot \frac{|H \cap T^+(e)|}{|H| - 1}.$$

- If $x \in H \cap T^*(e)$ then

$$\sum_{(b,x) \in H'_x} q_{b,x} \left( \frac{f_{b,x}(E \cup e) - f_{b,x}(E)}{1 - f_{b,x}(E)} \right) = \frac{p_x}{2} \cdot \frac{|H \cap T^-(e)| + |H \cap T^+(e)|}{|H| - 1}.$$

Summing over all $x \in H$, it follows that the second term in (6) equals the sum of the last three terms in (5). We now show that the first term in (6) equals the first term in (5):

$$\sum_{x \in L_e(H)} p_x = \sum_{x \in L_e(H)} \left( \sum_{(b,x) \in H'} q_{b,x} \right) = \sum_{(b,x) \in L'_e(H')} q_{b,x} = q\left( L'_e(H') \right),$$

where the first equality is by Lemma 4.4 and the second equality is by definition of $L_e(H)$ and $L'_e(H)$. It now follows that the score of each test at every state is equal in both algorithms.

*Proving 2:* We show that both algorithms stop at the same states. Algorithm 1 stops when $|H| = 1$ and Algorithm 2 stops when $H' = \emptyset$. Note that scenario-copy $(b,x)$ is covered exactly when $H = \{x\}$. As $H'$ consists of all compatible uncovered scenario-copies, it follows that $|H| = 1$ iff $H' = \emptyset$. $\square$

**Approximation ratio for Algorithm 2.** We will now prove

**Theorem 4.7.** *Algorithm 2 is an $O(r + \log m)$-approximation algorithm for ASR instance $\mathcal{J}$.*

The proof is very similar to the analysis in [24]. So we only provide an outline of the overall proof, while emphasizing the differences. For $k = 0, 1, \cdots$, define the following quantities:

- $A_k \subseteq M$ is the set of uncovered hypotheses in ALG at time $L \cdot 2^k$, and $a_k = q(A_k)$.
- $Y_k$ is the set of uncovered hypotheses in OPT at time $2^{k-1}$, and $y_k = q(Y_k)$.

Here $L = O(r + \log m)$. The key step is to show:

$$a_k \leq 0.2a_{k-1} + 3y_k, \qquad \text{for all } k \geq 1. \tag{7}$$

As shown in [24], this implies Theorem 4.7. In order to prove (7) we use the quantity:

$$Z := \sum_{t > L2^{k-1}}^{L2^k} \sum_{(E,H') \in R(t)} \max_{e \in U \setminus E} \left( \sum_{(b,x) \in L'_e(H')} q_{b,x} + \sum_{(b,x) \in H'} q_{b,x} \cdot \frac{f_{b,x}(e \cup E) - f_{b,x}(E)}{1 - f_{b,x}(E)} \right) \quad (8)$$

Above, $R(t)$ denotes the set of states $(E, H')$ that occur at time $t$ in ALG. (7) will be proved by separately lower and upper bounding $Z$.

**Lemma 4.8** ([24]). *We have $Z \geq L \cdot (a_k - 3y_k)/3$.*

The proof of this lower bound is identical to that in [24], although the definition of $L'_e(H')$ is different here. The proof of the upper bound however requires new ideas, as described next.

**Lemma 4.9.** *We have $Z \leq a_{k-1} \cdot (1 + \ln m + r + \log m)$.*

*Proof.* For any hypothesis $(b, x) \in A_{k-1}$ (i.e. uncovered in ALG by time $L2^{k-1}$) let $\sigma_{b,x}$ be the path traced by $(b, x)$ in ALG's decision tree, starting from time $2^{k-1}L$ and ending at $2^k L$ or when $(b, x)$ gets covered. Recall that for any $L2^{k-1} < t \leq L2^k$, any hypothesis in $H'$ for any state in $R(t)$ appears in $A_{k-1}$. So only hypotheses in $A_{k-1}$ can contribute to $Z$ and we rewrite (8) as:

$$Z = \sum_{(b,x) \in A_{k-1}} q_{b,x} \cdot \sum_{e \in \sigma_{b,x}} \left( \frac{f_{b,x}(e \cup E) - f_{b,x}(E)}{1 - f_{b,x}(E)} + \mathbb{1}[(b,x) \in L'_e(H')] \right)$$

$$\leq \sum_{(b,x) \in A_{k-1}} q_{b,x} \cdot \left( \sum_{e \in \sigma_{b,x}} \frac{f_{b,x}(e \cup E) - f_{b,x}(E)}{1 - f_{b,x}(E)} + \sum_{e \in \sigma_{b,x}} \mathbb{1}[(b,x) \in L'_e(H')] \right) \quad (9)$$

Above, for any $e \in \sigma_{b,x}$ we use $(E, H')$ to denote the state at which $e$ is selected.

Fix any hypothesis $(b, x) \in A_{k-1}$. For the first term, we use Lemma 4.10 below and the definition of $\epsilon$. This implies $\sum_{e \in \sigma_{b,x}} \frac{f_{b,x}(e \cup E) - f_{b,x}(E)}{1 - f_{b,x}(E)} \leq 1 + \ln \frac{1}{\epsilon} \leq 1 + \ln m$ as parameter $\epsilon \geq 1/m$ for $f_{b,x}$.

Now, we bound the second term by proving the inequality below:

$$\sum_{e \in \sigma_{b,x}} \mathbb{1}[(b,x) \in L'_e(H)] \leq r + \log m \quad (10)$$

To prove this inequality, consider hypotheses in $H'$. Now, if hypothesis $(b, x) \in L'_e(H)$ when ALG selects test $e$, then $x$ would be in $L_e(H)$. Suppose $L_e(H) = T^+(e) \cap H$; the other case is identical. Let $D_e(H) = T^-(e) \cap H$ and $S_e(H) = T^*(e) \cap H$. As $x \in L_e(H)$, it must be that path $\sigma_{b,x}$ follows the $+$ branch out of $e$. Also, the number of candidate hypotheses on this path after test $e$ is

$$|L_e(H)| + |S_e(H)| \leq \frac{|L_e(H)|}{2} + \frac{|D_e(H)|}{2} + |S_e(H)| = \frac{|H|}{2} + \frac{|S_e(H)|}{2} \leq \frac{|H|}{2} + \frac{r}{2}.$$

The first inequality uses the definition of $L_e(H)$ and the last inequality uses the bound of $r$ on the number of hypotheses with $*$ outcomes. Hence, each time that $(b, x) \in L'_e(H)$ along path $\sigma_{b,x}$, the number of candidate hypotheses changes as $|H_{new}| \leq \frac{1}{2}|H_{old}| + \frac{r}{2}$. This implies that after $\log_2 m$ such events, $|H| \leq r$. Let $\sigma'_{b,x}$ denote the portion of path $\sigma_{b,x}$ after $|H|$ drops below $r$. Note that each time $(b, x) \in L'_e(H)$ we have $L_e(H) \neq \emptyset$: so $|H|$ reduces by at least one after each such test $e$. Hence $\sum_{e \in \sigma'_{b,x}} \mathbb{1}[(b,x) \in L'_e(H)] \leq r$. As the portion of path $\sigma_{b,x}$ until $|H| \leq r$ contributes at most $\log_2 m$ to the left-hand-side in (10), the total is at most $r + \log_2 m$ as needed. $\square$

**Lemma 4.10.** *Let $f : 2^U \to [0, 1]$ be any monotone function with $f(\emptyset) = 0$ and $\epsilon = \min\{f(S \cup \{e\}) - f(S) : e \in U, S \subseteq U, f(S \cup \{e\}) - f(S) > 0\}$. Then, for any sequence $\emptyset = S_0 \subseteq S_1 \subseteq \cdots S_k \subseteq U$ of subsets, we have $\sum_{t=1}^{k} \frac{f(S_t) - f(S_{t-1})}{1 - f(S_{t-1})} \leq 1 + \ln \frac{1}{\epsilon}$.*

Setting $L = 15(1 + \ln m + r + \log_2 m)$ and applying Lemmas 4.8 and 4.9 completes the proof of (7) and hence Theorem 4.7.

**Tight Example:** our analysis above is tight, as shown by the following instance with $r = m$ where the algorithm's cost is $\Omega(m)$ times the optimum. The instance has $m = 6k$ scenarios, which are partitioned into $A = \{a_1, \cdots a_{3k}\}$, $B = \{b_1, \cdots, b_{3k-3}\}$ and $\{c, c', c''\}$. The probability of hypothesis $c$ is $1 - \epsilon$ and each of the other hypotheses has probability $\frac{\epsilon}{6k-1}$. We will use $\epsilon = k^{-3} \to 0$. We also have four types of tests (unspecified hypotheses have $*$ outcomes).

- $\alpha$-tests: for each $j \in [k]$, test $\alpha_j$ is $+$ on $\{c, c'\}$ and $-$ on $\{a_{3j-2}, a_{3j-1}, a_{3j}\}$.

- $\beta$-tests: for each $j \in [k-1]$, test $\beta_j$ is $+$ on $\{c, c'\}$ and $-$ on $\{b_{3j-2}, b_{3j-1}, b_{3j}\}$.

- $\gamma$-tests: for each pair $s, t \neq c$ of hypotheses there is a test that is $+$ on $s$ and $-$ on $t$.

- Test $\delta$ is $+$ on $A$ and $-$ on all other scenarios. Test $\delta'$ is $+$ on $\{c', c''\} \cup B$ and $-$ on all other scenarios.

**Bound on the Optimal Cost:** We first perform tests $\delta$ and $\delta'$. If both outcomes are $-$ then we identify hypothesis $c$ uniquely (which happens with probability $1 - \epsilon$). Otherwise, we continue to perform all the $\gamma$-tests which suffices to identify the realized hypothesis (this happens only with probability $\epsilon$). The expected cost is at most $2 + \epsilon \cdot m^2 = \mathcal{O}(1)$ using $\epsilon = 1/k^3$.

**Cost of our Algorithm:** We will only describe the sequence of tests under realized hypothesis $c$, which will provide a lower bound on the algorithm's cost. We claim that the algorithm will select tests $\alpha_1, \beta_1, \alpha_2, \beta_2, \cdots \alpha_{k-1}, \beta_{k-1}$ by suitable tie-breaking. We show this by induction. Consider the candidate hypotheses $H$ at any point in this sequence. Note that the alternating choice of $\alpha$ and $\beta$ tests implies that we will always have $c, c', c'' \in H$ and either $|H \cap A| = 3 + |H \cap B|$ or $|H \cap A| = |H \cap B|$. In either case, the "lighter" side of test $\delta$ is always $H \cap A$ (possibly by breaking ties) and the lighter side of test $\delta'$ is $H \cap (B \cup \{c', c''\})$. In particular, $c \notin L_e(H)$ for test $e \in \{\delta, \delta'\}$. The score of $e \in \{\delta, \delta'\}$ in Equation (5) is:

$$p(L_e(H)) \; + \; \frac{|T^+(e) \cap H|}{|H| - 1} \cdot p(T^-(e) \cap H) \; + \; \frac{|T^-(e) \cap H|}{|H| - 1} \cdot p(T^+(e) \cap H) \quad \leq \quad \epsilon + \frac{1}{2} + \epsilon.$$

Above we used the fact that hypothesis $c$ does not lie in the lighter side and it has probability $1 - \epsilon$ (so the total remaining probability is at most $\epsilon$). On the other hand, the score of all remaining $\alpha$ and $\beta$ tests will be equal (by symmetry) and has value at least $1 - \epsilon$ as $c$ lies in the lighter side of these tests. Finally, all $\gamma$-tests have a score of at most $\epsilon + \frac{1}{2}$. So the algorithm can choose any remaining $\alpha$ or $\beta$ test at this point, proving the inductive step. Thus the expected cost of our algorithm is at least $(1 - \epsilon)(2k - 2) = \Omega(m)$.

## 4.1 $\mathcal{O}(h + \log m)$-Approximation Algorithm

Here we observe that directly applying the ASR algorithm from [24] on instance $\mathcal{J}$ leads to this bound. This corresponds to changing $L'_e(H')$ in the score (6) to the smaller of the following sets:

$$\{(b, x) \in H' : r_{b,x}(e) = +\} \text{ and } \{(b, x) \in H' : r_{b,x}(e) = -\},$$

which corresponds to the smaller-cardinality part of $H'$.

**Theorem 4.11.** *There is an $\mathcal{O}(h + \log m)$-approximation algorithm for adaptive ODTN, where $h$ is the maximum number of noisy outcomes for a scenario.*

*Proof.* Consider the ASR instance $\mathcal{J}$. By applying the ASR algorithm, we obtain an $O(\log M + \log 1/\epsilon) = O(h + \log m)$ approximation ratio, because $M \leq 2^h \cdot m$ and $\epsilon \geq 1/m$. $\square$

**Corollary 4.11.1.** *There is an $\mathcal{O}(\min(h, r) + \log m)$-approximation algorithm for adaptive ODTN, where $h$ is the maximum number of noisy outcomes per scenario, and $r$ is the maximum number of noisy outcomes per test.*

We note that our analysis is tight (up to constant factors).

# 5 Adaptive Algorithm with Very Few Known Outcomes

In this section we consider instances with a very large number of noisy outcomes, and obtain an approximation algorithm that relies on different ideas. Formally, we define an $\alpha$-**sparse** ($\alpha \leq 1/2$) instance as follows. There is a constant $C$ such that $\max\{|T^+(e)|, |T^-(e)|\} \leq C \cdot m^\alpha$ for all tests $e \in U$. Our main result here is:

**Theorem 5.1.** *There is an adaptive $O(\log m)$-approximation algorithm for ODT with noise on any $\alpha \leq \frac{1}{2}$ sparse instance.*

We first make some simple observations related to $\alpha$-sparse instances.

**Proposition 1.** *The optimal value $OPT \geq \Omega(m^{1-\alpha})$.*

*Proof.* By definition of $\alpha$-sparse instances, the maximum number of candidate hypotheses that can be eliminated after performing a single test is $m^\alpha$. As we need to eliminate $m - 1$ hypotheses irrespective of the realized hypothesis $\bar{x}$, we need to perform at least $\frac{m-1}{m^\alpha} = \Omega(m^{1-\alpha})$ tests under every $\bar{x}$. $\square$

**Proposition 2.** *Consider any $W \subseteq U$ and $X \subseteq [m]$. For any $y \in X$, $\kappa(y) = |\{e \in W : r_y(e) \neq *\}|$ denotes the number of tests in $W$ for which $y$ has a $\pm 1$ outcome. Then, the number of hypotheses in $X$ with $\kappa(y) > |W|/2$ is at most $2Cm^\alpha$.*

*Proof.* Let $X' = \{y \in X : \kappa(y) > |W|/2\}$. Then:

$$|X'| \cdot \frac{|W|}{2} < \sum_{y \in X} \kappa(y) = \sum_{e \in W} |\{y \in X : r_y(e) \neq *\}| \leq |W| \cdot Cm^\alpha.$$

Rearraging, we get $|X'| \leq 2Cm^\alpha$ as needed. $\square$

*Main Idea.* Consider a decision tree obtained by the following naive algorithm: when $A$ is the current subset of "alive" scenarios, we choose $T$ s.t. $|T^+ \cap A| + |T^- \cap A|$ is maximized. In general, this tree can have very high cost compared to OPT, so we truncate it to reduce the cost to $O(\log m \cdot OPT)$. Meanwhile, our truncated tree is guaranteed to rule out all but a "small" number ($O(\sqrt{m})$) of scenarios, hence it won't be too costly to perform a "brute-force" search on the remaining ones.

## 5.1 Relation to Stochastic Set Cover

We now establish a crucial relation to the stochastic set cover (SSC) problem [26, 22] and also state a useful result on SSC. This forms the basis of our algorithm.

An instance of SSC consists of a groundset $V$ and $k$ stochastic sets $S_1, \cdots S_k$ each of which is a subset of $V$. The distribution of each set $S_i$ is known to the algorithm and the distributions of different sets are independent of each other. The instantiation of each set is only known after it is selected. The goal is to find an adaptive policy that minimizes the expected number of sets to cover $V$. A natural adaptive greedy algorithm is known to be an $O(\log m)$-approximation where $m = |V|$ [26]. At any point in this policy, if $A \subseteq V$ denotes the uncovered elements then we choose the set $S_{i^*}$ that maximizes the expected number of new elements covered, i.e. $i^* = \arg\max_{i=1}^k \mathbb{E}[|S_i \cap A|]$. At any such step, we call a set $\beta$-greedy if it has expected coverage at least a $1/\beta$ fraction of the maximum. We need an extension of this result that involves picking $\beta$-greedy sets at just some constant fraction of the steps. Formally, call an SSC policy $(\beta, \rho)$ greedy if for every $t \geq 1$, at least $t/\rho$ steps among the first $t$ steps involve picking a $\beta$-greedy set. By modifying the analysis in [22] slightly, we obtain:

**Theorem 5.2.** *Any $(\beta, \rho)$ greedy policy for stochastic set cover is an $O(\beta\rho \log m)$-approximation.*

We now derive a lower bound on the optimal ODTN value in terms of certain SSC instances. For any $x \in [m]$, let $SSC(x)$ denote the stochastic set cover instance with groundset $V = [m] \setminus \{x\}$ (i.e. all hypotheses other than $x$) and sets $U$ (i.e. all tests) where

$$S_e(x) = \begin{cases} T^+(e) \text{ with prob. } 1 & \text{if } x \in T^-(e) \\ T^-(e) \text{ with prob. } 1 & \text{if } x \in T^+(e) \\ T^-(e) \text{ or } T^+(e) \text{ with prob. } \frac{1}{2} \text{ each} & \text{if } x \in T^*(e) \end{cases}, \qquad \forall e \in U.$$

**Lemma 5.3.** $OPT \geq \sum_{x \in [m]} \pi_x \cdot OPT_{SSC(x)}$.

*Proof.* Consider any feasible decision tree $\mathcal{T}$ for the ODTN instance and any hypothesis $x \in [m]$. If we *condition* on $\bar{x} = x$ then $\mathcal{T}$ corresponds to a feasible adaptive policy for $SSC(x)$. This is because (i) for any outcome-vector $b \sim x$, the tests performed in $\mathcal{T}$ must rule-out all the hypotheses $[m] \setminus x$ and (ii) the hypotheses ruled-out by any test $e$ (conditioned on $\bar{x} = x$) is a random subset that has the same distribution as $S_e(x)$. Formally, if $P_{b,x}$ denotes the path traced in $\mathcal{T}$ under test outcomes $b \sim x$ then this policy for $SSC(x)$ has cost $\sum_{b \sim x} 2^{-h_x} \cdot |P_{b,x}|$ as $\Pr[\text{observing outcomes } b | \bar{x} = x] = 2^{-h_x}$. So $OPT_{SSC(x)} \leq \sum_{b \sim x} 2^{-h_x} \cdot |P_{b,x}|$. Taking expectations over $x \in [m]$ the lemma follows. $\square$

## 5.2 A Low-Cost Membership Oracle

One subroutine in our algorithm is an "oracle" called "Member", which takes a (small) subset $Z \subseteq [m]$ as input, and decides whether $\bar{x} \in Z$.

1. Initialize $Z' \leftarrow Z$.

2. While $|Z'| \geq 2$, do:

   i. Perform any test $e \in U$ with both $T^+(e) \cap Z', T^-(e) \cap Z' \neq \emptyset$,
   ii. Let $R$ be the set of scenarios ruled out, let $Z' \leftarrow Z' \setminus R$.

3. When $|Z'| = 1$, let $Z' = \{z\}$.          % Determine whether $\bar{x} = z$.

   (a) $Y \leftarrow [m] \setminus \{z\}$, $k \leftarrow 0$,
   (b) while $Y \neq \emptyset$ and $k \leq 4 \log m$, do:
   > choose any test $e \in U$ with $z \notin T^*(e)$, suppose that $z \in T^+(e)$.[4] If the outcome of test $e$ is inconsistent with $z$, then declare "$\bar{x} \notin Z$" and stop; otherwise let $Y \leftarrow Y \setminus T^-(e)$ and $k \leftarrow k + 1$.
   (c) Let $W \subseteq U$ denote the tests performed in step 3b and
   $$S = \{y \in [m] : r_y(e) = r_z(e) \text{ for at least } 2 \log m \text{ tests } e \in W\}.$$
   (d) For each $y \in S$, perform any test that deterministically separates $y$ and $z$.
   (e) If we never received an outcome inconsistent with $z$ in step 3d, then declare "$\bar{x} = z$"; otherwise declare "$\bar{x} \notin Z$".

One caveat in step 3b is that it may happen that there are not $4 \log m$ many tests $e$ with $z \notin T^*(e)$. However, in this case, the algorithm must have already exhausted *all* tests with $z \notin T^*(e)$ and because $z$ is still consistent with all observed outcomes, we know that $\bar{x} = z$.

**Lemma 5.4.** *If $\bar{x} \in Z$, then Member($Z$) declares $\bar{x} = z$ with probability one; otherwise, Member($Z$) declares $\bar{x} \notin Z$ with probability $1 - m^{-2}$. Moreover, the expected cost of Member($Z$) is $O(|Z| + m^{\alpha})$.*

*Proof.* If $\bar{x} \in Z$ then it is clear that Member($Z$) declares $\bar{x} = z$. Now consider the case $\bar{x} \notin Z$. Recall that $z \in Z$ denote the unique hypothesis that is still compatible after step 2.

Case 1   If $\bar{x} \notin S$ then we have $\bar{x} \in T^*(e)$ for at least $2 \log m$ tests $e \in W$. As $z$ has a deterministic outcome for each test in $W$, the probability that all outcomes in $W$ are consistent with $z$ is at most $m^{-2}$. So with probability $1 - m^{-2}$ we must have observed that $z$ is not compatible in step 3b and declare $\bar{x} \notin Z$.

Case 2   If $\bar{x} \in S$ then we will identify correctly that $\bar{x} \neq z$ in step 3c as one of these tests separates $\bar{x}$ and $z$ deterministically. So in this case we will always declare $\bar{x} \notin Z$.

In order to bound the cost, note that the number of tests performed are: $|Z|$ in step 2, $4 \log m$ in step 3b and $|S|$ in step 3d. The key step is to bound $|S|$, for which we use Proposition 2 with tests $W$ and hypotheses $X = [m]$. In the notation of Proposition 2, $S = \{y \in X : \kappa(y) > |W|/2\}$ as $|W| = 4 \log m$. It now follows that $|S| \le 2Cm^\alpha$. Hence the total number of tests is $|Z| + 4 \log m + |S| = O(|Z| + m^\alpha)$. $\qquad\square$

## 5.3 The Main Algorithm

Now define the main algorithm $\mathcal{A}_0$:

1. Initialize: compatible hypotheses $A \leftarrow [m]$, weights $w_x = 0$ for $x \in A$, number of tests $t \leftarrow 0$.

2. While $|A| \ge 2$,

   (a) If $t$ is a power of 2, let $Z \subseteq A$ be the subset of $2Cm^\alpha$ scenarios with lowest $w_x$. Invoke Member($Z$): if it identifies some $Z$-hypothesis as $\bar{x}$ then stop.

   (b) Else:
      i. Perform test $e \in U$ maximizing $\frac{1}{2}(|T^+(e) \cap A| + |T^-(e) \cap A|)$.
      ii. Update the weights: $w_x \leftarrow w_x + 1$ for each for each $x \in T^*(e)$.
      iii. Remove incompatible hypotheses from $A$ (based on the test $e$ outcome).
      iv. $t \leftarrow t + 1$.

3. Output the unique hypothesis in $A$ as $\bar{x}$.

Using Lemma 5.4, it is clear that $\bar{x}$ is identified correctly with probability $1 - m^{-2}$. We now analyze the cost. Note that the membership oracle is invoked $O(\log m)$ times as the total number $t$ of tests used is always at most $m$. Moreover, each time it is invoked on $O(m^\alpha)$ scenarios. So the total number of tests due to step 2a is $O(m^\alpha \log m)$. In the rest of this section, we will bound the expected cost due to step 2b and ignore the cost of performing tests due to step 2a.

**Truncated Decision Tree.** Let $\mathcal{A}$ denote the decision tree corresponding to our algorithm. We only consider tests that correspond to step 2b. For any expanded hypothesis $(b, x) \in M$ let $P_{b,x}$ denote the path traced in $\mathcal{A}$. We will work with a truncated decision tree $\bar{\mathcal{A}}$, defined as follows.

Fix any $(b, x) \in M$. Let $\theta(t)$ denote the fraction of the first $t$ tests in $P_{b,x}$ that are $*$-tests for $x$.[5]

$$\text{Define } t_{b,x} = \max\left\{ t \in \{2^0, 2^1, \cdots, 2^{\log m}\} \ : \ \theta(t') \ge \frac{1}{\rho} \text{ for all } t' \le t \right\}. \tag{11}$$

We let $\rho = 4$. If $t_{b,x} > |P_{b,x}|$ then reset $t_{b,x} = |P_{b,x}|$.

$\bar{\mathcal{A}}$ is the subtree of $\mathcal{A}$ consisting of the first $t_{b,x}$ tests along path $P_{b,x}$, for each $(b, x) \in M$.

**Relating costs of $\mathcal{A}$ and $\bar{\mathcal{A}}$.** We now show that the expected cost of $\mathcal{A}$ is at most twice that of $\bar{\mathcal{A}}$. We will show that for each $(b, x) \in M$, the number of tests performed $|P_{b,x}| \le 2t_{b,x}$. Again, we only count tests from step 2b. We only need to consider the case that $t_{b,x} < |P_{b,x}|/2$. Let $t'_{b,x} = 2t_{b,x}$ which is a power-of-2. By (11) we know that there is some $t_{b,x} < k \le t'_{b,x}$ with $\theta(k) < 1/\rho$. Hence $\theta(t'_{b,x}) < \frac{2}{\rho} < \frac{1}{2}$. Consider the point in the algorithm after performing the first $t'_{b,x}$ tests (call them $W$) on $P_{b,x}$. Let $X$ be the compatible hypotheses after the $t'_{b,x}$-th test on $P_{b,x}$. Because $\theta(t'_{b,x}) < 1/2$, at most $|W|/2$ tests in $W$ are $*$-tests for hypothesis $x$: in other words the weight $w_x \le |W|/2$ at this point in the algorithm. Let $X' = \{y \in X : W \text{ has at most } \frac{|W|}{2} *\text{-tests for } y\}$. Using Proposition 2 with $W$ and $X$, it follows that $|X'| \le 2Cm^\alpha$. Hence the number of hypotheses $y \in X$ with $w_y \ge |W|/2$ is at most $2Cm^\alpha$, and so $x \in Z$ (recall that $Z$ consists of $2Cm^\alpha$ hypotheses with the lowest weight).

**Bounding cost of $\bar{\mathcal{A}}$.** Here we use the relation to stochastic set cover. Recall the instances $SSC(x)$ for $x \in [m]$. A key observation is:

**Proposition 3.** *Consider step 2b in the main algorithm, where $A$ denotes the set of compatible hypotheses and $e$ is the chosen test. For any $x \in T^*(e)$,*

$$\frac{1}{2}\left(|T^+(e) \cap A| + |T^-(e) \cap A|\right) = \mathbb{E}[|S_e(x) \cap (A \setminus x)|] \geq \frac{1}{2} \cdot \max_{d \in U} \mathbb{E}[|S_d(x) \cap (A \setminus x)|].$$

*Proof.* This follows easily by the choice of test $e$ in step 2b. Suppose (for contradiction) there is some $d \in U$ with $\mathbb{E}[|S_d(x) \cap (A \setminus x)|] > 2 \cdot \mathbb{E}[|S_e(x) \cap (A \setminus x)|]$.

- If $x \in T^*(d)$ then $\mathbb{E}[|S_d(x) \cap (A \setminus x)|] = \frac{1}{2}\left(|T^+(d) \cap A| + |T^-(d) \cap A|\right) > |T^+(e) \cap A| + |T^-(e) \cap A|$, which violates the choice of $e$.

- If $x \in T^+(d)$ then $\mathbb{E}[|S_d(x) \cap (A \setminus x)|] = |T^-(d) \cap A| > |T^+(e) \cap A| + |T^-(e) \cap A|$, which again violates the choice of $e$.

In either case, we obtain a contradiction. $\square$

Fix any hypothesis $x \in [m]$ and consider decision tree $\bar{\mathcal{A}}_x$ obtained by *conditioning* $\bar{\mathcal{A}}$ on $\bar{x} = x$. The truncation (11) and Proposition 3 together imply that $\bar{\mathcal{A}}_x$ is a $(2, \rho)$ greedy policy for $SSC(x)$. Now, using Theorem 5.2, the expected cost of $\bar{\mathcal{A}}_x$ is $O(\log m) \cdot OPT_{SSC(x)}$. Taking expectations over $x \in [m]$, the expected cost of $\bar{\mathcal{A}}$ is $O(\log m) \sum_{x=1}^m \pi_x \cdot OPT_{SSC(x)}$, which is $O(\log m) \cdot OPT$ by Lemma 5.3.

Combined with the fact that the cost of $\mathcal{A}$ is at most twice that of $\bar{\mathcal{A}}$, the expected number of tests due to step 2b in the main algorithm is $O(\log m) \cdot OPT$. Adding the contribution from step 2a, we obtain an expected cost of

$$O(\log m) \cdot (m^\alpha + OPT) \leq_{(\text{as } \alpha < \frac{1}{2})} O(\log m) \cdot (m^{1-\alpha} + OPT) \leq_{(\text{Prop. 1})} O(\log m) \cdot OPT.$$

This completes the proof of Theorem 5.1.

# 6 Extensions

**Instances that are not perfectly identifiable.** We have assumed that for every pair $x, y$ of hypotheses, there is some test that distinguishes them deterministically. Without this assumption, we can still obtain similar results by slightly changing the stopping criterion as follows. Define a *similarity graph* $G$ on $m$ nodes (corresponding to hypotheses) with an edge $(x, y)$ if there is *no* test separating $x$ and $y$ deterministically. For each $x \in [m]$, let $D_x$ be the star centered at $x$. We now want a policy using minimum number of tests that identifies a star containing $\bar{x}$. In the noise-less case, this problem has been studied in [23, 9] and an adaptive $O(d + \log m)$-approximation algorithm is known [24] where $d = \max_{i=1}^m |D_i|$. For us, $d = 1 + \text{max-degree}(G)$. Theorems 3.2 and 4.11.1 can be extended to obtain approximation ratios of $O(d \log m)$ and $O(d + \min(h, r) + \log m)$ respectively.

**Non-binary outcomes.** We can also handle tests with an arbitrary set $\Sigma$ of outcomes (instead of $\pm 1$). This requires extending the outcomes $b$ to be in $\Sigma^U$ and applying this change to the definitions of sets $T_{b,x}$ (1) and submodular function $f_{b,x}$ (2).

For the non-adaptive version, we will apply the approach using submodular ranking using different submodular functions $f_{b,x}$. In particular, for each $(b, x) \in M$ and region $D_i$, we define a submodular function:

$$f_{b,x}^i(S) = |\bigcup_{e \in S} T_{b,x}(e) \cap \overline{D_i}| \cdot \frac{1}{m - |D_i|}, \qquad \forall S \subseteq U,$$

where $\overline{D_i} = [m] \setminus D_i$. Assuming $\bar{x} = x$ and test-outcomes are given by $b$, we know $x \in D_i$ after tests $S$ if and only if $f_{b,x}^i(S) = 1$. We define $f_{b,x}$ to be the "OR combination" of functions $f_{b,x}^i$ where $x \in D_i$. As in [24],

$$f_{b,x}(S) = \Pi_{i:x \in D_i}(1 - f_{b,x}^i(S)), \qquad \forall S \subseteq U.$$

Crucially, this OR combination ensures that $f_{b,x}(S) = 1$ iff some $f^i_{b,x}(S) = 1$; see [18]. The marginal increment parameter $\epsilon = m^{-d}$ as $|\{i : x \in D_i\} \leq d$ for all $x \in [m]$. The non-adaptive algorithm is then identical to that in § 3 and we obtain an $O(\log \frac{1}{\epsilon}) = O(d \log m)$ approximation ratio.

For the adaptive version, following the noise-less algorithm in [24], we run a two-phase algorithm. In the first phase, we identify some subset $N \subseteq [m]$ containing $\bar{x}$ with $|N| \leq d$. This can be done using the algorithm in § 4 with the following submodular function for each $(b, x) \in M$.

$$ f_{b,x}(S) == |\bigcup_{e \in S} T_{b,x}(e)| \cdot \frac{1}{m - d}, \qquad \forall S \subseteq U. $$

The expected cost of the resulting algorithm is $O(\min(r, h) + \log m) \cdot OPT$ using an identical analysis. Then, in the second phase, we run a simple splitting algorithm that iteratively selects any test that splits the current set $H$ of candidate hypotheses, until $H$ is contained in some region. The expected cost of this phase is at most $d \cdot OPT$. Combining both phases, we obtain an $O(d + \min(h, r) + \log m)$-approximation algorithm.

**Non-uniform noise distribution.** Our results extend directly to the case where each noisy outcome has a different probability of being $\pm 1$. Suppose that the probability of every noisy outcome is between $\delta$ and $1 - \delta$. Then Theorems 3.2 and 4.11.1 continue to hold (irrespective of $\delta$), and Theorem 5.1 holds with a slightly worse $O(\frac{1}{\delta} \log m)$ approximation ratio.

## 7 Experiments

We implemented our algorithms, and performed experiments on real-world and synthetic data sets. We compared our algorithms' cost (expected number of tests) with an information theoretic lower bound on the optimal cost and show that the difference is negligible. Thus, despite our logarithmic approximation ratios, the practical performance can be much better.

**Chemicals with Unknown Test Outcomes** One natural application of ODT is identifying chemical or biological materials. We considered a data set called WISER[6], which includes 400+ chemicals (hypothesis) and 78 binary tests. Every chemical has either positive, negative or unknown result on each test. To ensure every pair of chemical can be distinguished, we removed the chemicals that are not identifiable from each other to be left with 255 chemicals.

**Random Binary Classifiers with Margin Error** We construct a dataset containing 100 two-dimensional points, by picking each of their attributes uniformly in $[-1000, 1000]$. We also choose 2000 random triples $(a, b, c)$ to form linear classifiers $\frac{ax+by}{\sqrt{a^2+b^2}} + c \leq 0$, where $a, b \leftarrow N(0, 1)$ and $c \leftarrow U(-1000, 1000)$. The point labels are binary and we introduce noisy outcomes based on the distance of each point to a classifier. Specifically, for each threshold $d \in \{0, 5, 10, 20, 30\}$ we define dataset CL-$d$ that has a noisy outcome for any classifier-point pair where the distance of the point to the boundary of the classifier is smaller than $d$. In order to ensure that the instances are perfectly identifiable, we remove "equivalent" classifiers and we are left with 234 classifiers.

**Distributions** For the distribution over the hypotheses we have considered permutations of power law distribution ($\Pr[X = x; \alpha] = \beta x^\alpha$) for $a = 0, -0.5$ and 1. Note that, $\alpha = 0$ corresponds to uniform distribution. To be able to compare the results across different classifiers' datasets meaningfully. We have considered the same permutation in each distribution.

**Algorithms and Results** The following algorithms is implemented on Mac OS with 16GB RAM in Python 3 : the adaptive $O(r + \log m)$-approximation (ODTN$_r$), the adaptive $O(h + \log m)$-approximation (ODTN$_h$), the non-adaptive $O(\log m)$-approximation (Non-Adap) and a slightly adaptive version of Non-Adap (Low-Adap). Algorithm Low-Adap considers the same sequence of tests as Non-Adap while (adaptively) skipping non-informative tests based on observed outcomes. To take randomness of unknown outcomes into account, we run each algorithm 100 times. Tables 1,

Tables 3 and Tables 5 show the expected costs of different algorithms on all datasets when the parameter $\alpha$ in the distribution over hypothesis is $0, -0.5$ and $-1$ correspondingly. These tables also report values of an information theoretic lower bound (the entropy) on the optimal cost (Low-BND). We also report the sample standard deviation of all algorithms in Tables 2, Tables 4 and Tables 6. Since the approximation ratio of some of our algorithms are dependent on maximum number of unknown outcomes per hypothesis (h) and maximum number of unknown outcomes per test (r), we also have included these parameters as well as their average values . From the results We can see that $ODTN_r$ consistently outperforms the other algorithms in every distribution and is very close to the lower bound.

| Data ⟍ Algorithm | Wiser | Classifer-0 | Classifer-5 | Classifer-10 | Classifer-20 | Classifer-30 |
|---|---|---|---|---|---|---|
| **Low-BND** | **7.994** | **7.870** | **7.870** | **7.870** | **7.870** | **7.870** |
| $ODTN_r$ | 8.357 | 7.910 | 7.927 | 7.915 | 7.962 | 8.000 |
| $ODTN_h$ | 9.707 | 7.910 | 7.979 | 8.211 | 8.671 | 8.729 |
| Non-Adap | 11.568 | 9.731 | 9.831 | 9.941 | 9.996 | 10.204 |
| Low-Adap | 9.152 | 8.619 | 8.517 | 8.777 | 8.692 | 8.803 |

Table 1: Cost of Different Algorithms for $\alpha = 0$ (Uniform Distribution).

| Data ⟍ Algorithm | Wiser | Classifer-0 | Classifer-5 | Classifer-10 | Classifer-20 | Classifer-30 |
|---|---|---|---|---|---|---|
| $ODTN_r$ | 0.008 | 0 | 0 | 0.002 | 0.003 | 0.006 |
| $ODTN_h$ | 0.01 | 0 | 0 | 0 | 0.004 | 0.01 |
| Non-Adap | 1.463 | 0.937 | 1.047 | 1.092 | 1.056 | 1.158 |
| Low-Adap | 0.0317 | 0.0685 | 0.0541 | 0.0760 | 0.0206 | 0.0550 |

Table 2: Standard Deviation of Different Algorithms for $\alpha = 0$ (Uniform Distribution).

| Data ⟍ Algorithm | Wiser | Classifer-0 | Classifer-5 | Classifer-10 | Classifer-20 | Classifer-30 |
|---|---|---|---|---|---|---|
| Low-BND | 7.702 | 7.582 | 7.582 | 7.582 | 7.582 | 7.582 |
| $ODTN_r$ | 8.177 | 7.757 | 7.780 | 7.789 | 7.831 | 7.900 |
| $ODTN_h$ | 9.306 | 7.757 | 7.829 | 8.076 | 8.497 | 8.452 |
| Non-Adap | 11.998 | 9.504 | 9.500 | 9.694 | 9.826 | 9.934 |
| Low-Adap | 8.096 | 7.837 | 7.565 | 7.674 | 8.072 | 8.310 |

Table 3: Cost of Different Algorithms for $\alpha = -0.5$.

| Data ⟍ Algorithm | Wiser | Classifer-0 | Classifer-5 | Classifer-10 | Classifer-20 | Classifer-30 |
|---|---|---|---|---|---|---|
| $ODTN_r$ | 0.008 | 0 | 0 | 0.002 | 0.005 | 0.005 |
| $ODTN_h$ | 0.008 | 0 | 0.003 | 0.004 | 0.008 | 0.007 |

Table 4: Standard Deviation of Different Algorithms for $\alpha = -0.5$.

| Data \ Algorithm | Wiser | Classifer-0 | Classifer-5 | Classifer-10 | Classifer-20 | Classifer-30 |
|---|---|---|---|---|---|---|
| Low-BND | 6.218 | 6.136 | 6.136 | 6.136 | 6.136 | 6.136 |
| $ODTN_r$ | 7.367 | 6.998 | 7.121 | 7.150 | 7.299 | 7.357 |
| $ODTN_h$ | 8.566 | 6.998 | 7.134 | 7.313 | 7.637 | 7.915 |
| Non-Adap | 11.976 | 9.598 | 9.672 | 9.824 | 10.159 | 10.277 |
| Low-Adap | 9.072 | 8.453 | 8.344 | 8.609 | 8.683 | 8.541 |

Table 5: Cost of Different Algorithms for $\alpha = -1$.

| Data \ Algorithm | Wiser | Classifer-0 | Classifer-5 | Classifer-10 | Classifer-20 | Classifer-30 |
|---|---|---|---|---|---|---|
| $ODTN_r$ | 0.004 | 0 | 0.001 | 0.001 | 0.003 | 0.003 |
| $ODTN_h$ | 0.005 | 0 | 0.001 | 0.003 | 0.005 | 0.004 |

Table 6: Standard Deviation of Different Algorithms for $\alpha = -1$.

| Data \ Parameters | Wiser | Classifer-0 | Classifer-5 | Classifer-10 | Classifer-20 | Classifer-30 |
|---|---|---|---|---|---|---|
| r | 245 | 0 | 5 | 7 | 12 | 13 |
| Avg-r | 30.690 | 0 | 1.12 | 2.21 | 4.43 | 6.54 |
| h | 45 | 0 | 3 | 6 | 8 | 8 |
| Avg-h | 9.39 | 0 | 0.48 | 0.94 | 1.89 | 2.79 |

Table 7: Maximum and Average Number of Stars per Hypothesis and per Test in Different Datasets.

## Footnotes

[1]We consider binary test outcomes only for simplicity: our results also hold for finitely many outcomes.

[2]There are also instances where the relative gap between the best adaptive and non-adaptive policies is $\tilde{\Omega}(m)$.

[3]The paper [15] states the approximation ratio as $O(\log 1/p_{min})$ because it relied on an erroneous claim in [13]. The correct approximation ratio, based on [30, 14], is $O(\log^2 1/p_{min})$.

[33], who proved that the natural greedy algorithm is a $(1 + \ln \frac{1}{\epsilon})$-approximation algorithm, where $\epsilon$ is the minimal positive marginal increment of the function. [3] obtained an $O(\log \frac{1}{\epsilon})$-approximation algorithm for the submodular ranking problem, that involves simultaneously covering multiple submodular functions; [22] extended this result to also handle costs. [24] studied an adaptive version of the submodular ranking problem. We utilize results/techniques from these papers.

[4]The case $z \in T^-(e)$ is identical.

[5] Only tests from step 2b are counted.

[6]https://wiser.nlm.nih.gov