[Reviews · NeurIPS 2019]

Reviewer 1



The setup is original and I see high value in the persistent-noise assumption worked out by the authors. I do have one main question to the authors and while I recommend this paper to be accepted based on significance and appearance of correctness, I do expect a very strong answer on this point for the score to remain high after rebuttal phase. The authors state in their experiment: "To ensure every pair of chemicals can be distinguished, we removed the chemicals that are not identifiable from each other." Well, for significance of the present work, we also need to know how the algorithms are going to behave in the worst-case if there are symmetries and this kind of preprocessing step is omitted. Note that the user would be happy with being presented a set of hypotheses and a certificate that no further test is available to distinguish among them. But I do want this certificate, I do want the analysis to be able to give me the expected time to get it, and I do want the experimental results in the paper extended to the case without the preprocessing that reduce solution sets to singletons by altering the real data set (which makes it a synthetic data set optimized to validating the proposal). UPDATE: Author response read. Based on good responses, score of this reviewer to remain as Clear accept.

Reviewer 2



This paper considers challenging settings for ODT problems with noisy outcomes. The authors consider several new settings such as unknown number of hypotheses and persistent noisy models. The authors design new adaptive and non-adaptive algorithms to achieve (nearly) best possible approximation ratio. I have to say that I don't work on approximate algorithms. The results look interesting to me, but I am not at a good position to judge the novelty.

Reviewer 3



I comment here on the long version of the paper, which is actually easier to read than the short version. Title: it oversells the scope of the results, since the only type of "noisy" outcomes * handled thoroughly in the paper are erasures with equal probability to be +/-. Section 3: - the proof of Lemma 3.4 shows that G_E(e*) > S/4, but the statement of of Lemma 3.4 claims that G_E(e*) > S/2. - the proof of Lemma 3.5 states that y \noin \cup T_{b,x}(e) f_{b,x}(E): I guess that f_{b,x}(E) should not be there. Section 4: - The equivalence between both algorithms is interesting, but it basically boils down in showing that the way the * entries are treated is the same for both instances. It would be good to clearly identify the alternative algorithm ODTN_h in Section 4.1 (only ODTN_r is actually introduced in Section 4). Section 5: - in Proposition 2, how do you get that the number of hypotheses in X' is at most 2Cm^\alpha and not 4Cm^\alpha, since |T^{+}(e) \cup T^{-}(e)| \leq 2 \max{ |T^{+}(e)|, |T^{-}(e)| } \leq 2Cm^\alpha ? REVISION: Comment on Section 7 deleted after authors' feedback, which clarified my question, indeed r = 245 and h = 45 are correct. Other comments remain unchanged.

Reviewer 4



The paper is well-written, carefully done, and a good contribution to the area. See answers to question 1. UPDATE: Read author feedback, which did a good job of answering the questions/concerns raised by the other reviewers. I'm still scoring this paper as a clear accept.

[Author Response · NeurIPS 2019]

We thank all the reviewers for a thorough reading and their helpful comments. Our responses are below. We refer to sections/pages etc in the full version of our submission.

**Instances that are not perfectly identifiable (Reviewer 1).** As stated in §6, our results can be extended to the case where not all pairs of hypotheses can be distinguished. There is, however, some loss in the performance guarantee, which now also depends on the maximum degree of the *similarity graph* $G$ (defined in §6, first paragraph). Graph $G$ contains an edge for every pair of hypotheses that are not identifiable from each other. Let $d = 1+$ max-degree$(G)$. Note that $G$ is empty (and $d = 1$) for perfectly-identifiable instances (assumed in §1-5).

*Example:* Consider hypotheses $\{1, 2, \cdots m\}$ and $m$ tests where the $i^{th}$ test (for any $i = 1, \cdots m$) has (a) outcome $-$ for hypotheses $\{1, \cdots i - 1\}$, (b) outcome $*$ for hypothesis $i$ and (c) outcome $+$ for hypotheses $\{i + 1, \cdots m\}$. The similarity graph $G$ here is a line with edges $(i, i + 1)$ for all $i = 1, \cdots m - 1$. So $d = 3$ for this instance.

Our current description in §6 gives a policy that stops when the compatible hypotheses $H$ is a subset of any star in $G$, which we call the *neighborhood stopping criterion*. (The paragraph on "Non-binary outcomes" was unfortunately misplaced.) The description in pages 15-16 outlines how to obtain a non-adaptive $O(d \cdot \log m)$-approximation and an adaptive $O(d + \min(h, r) + \log m)$-approximation for neighborhood stopping. Note that this matches the results stated in Theorem 3.2 and Corollary 4.11.1 where $d = 1$. In fact, our adaptive algorithm's guarantee is stronger: the cost of our algorithm is at most $O(\min(h, r) + \log m) \cdot OPT + d$.

The stopping criterion suggested by Reviewer 1 requires the compatible hypotheses $H$ to be a clique in $G$ (so there is no further test to distinguish between them). We call this the *clique stopping criterion*; note that this is a stricter requirement than neighborhood stopping. Our adaptive algorithm can be easily extended to this criterion. Note that $|H| \leq d$ at the end of our policy for neighborhood stopping. We then continue performing tests that distinguish within $H$ until $H$ is completely indistinguishable (i.e., a clique in $G$). The number of additional tests is at most $d$ (each test reduces $|H|$ by at least one), which does not affect our worst-case guarantees.

We also tested our algorithms on the WISER dataset (without preprocessing) using both the neighborhood and clique stopping criteria and the results are reported below (for uniform distribution). The resulting similarity graph has $d = 54$ and the number of hypotheses $m = 414$. The preprocessed instance (reported in the submission and reproduced in the first column below) has a smaller set of hypotheses, chosen so that they are perfectly identifiable (we used a greedy rule that iteratively drops the highest-degree hypothesis in $G$). While we agree that preprocessing can change the objective in an unpredictable way, we think that it still preserves some structure of the original dataset.

| Algorithm | Wiser (preprocessed) | Wiser (neighborhood stopping) | Wiser (clique stopping) |
|---|---|---|---|
| # Hypotheses ($m$) | 255 | 414 | 414 |
| ODTN$_r$ | 8.357 | 11.163 | 11.817 |
| ODTN$_h$ | 9.707 | 11.908 | 12.506 |
| Non-Adap | 11.568 | 16.995 | 21.281 |
| Low-Adap | 9.152 | 16.983 | 20.559 |

In summary, we present extensions of our results to output a set of scenarios along with a witness that no further distinguishing between any pair in this set is possible (since we return a clique in the similarity graph). In doing this, we achieve a generalized performance ratio of $O(d + \min(h, r) + \log m)$ in the adaptive setting, and promising results on preliminary experiments with the WISER dataset.

**Arbitrary probabilities in the noise model (Reviewer 4).** As stated in §6 (last paragraph), our results continue to hold in the setting where each noisy outcome has a different (arbitrary) probability to be $+/-$. Theorem 3.2 and Corollary 4.11.1 are unchanged. The approximation ratio in Theorem 5.1 increases by a factor of $\frac{1}{\delta}$ where $\delta > 0$ is the minimum probability of any noisy outcome (assumed to be $\frac{1}{2}$ in §1-5). We decided to focus on the simpler (but representative) case of uniform $\pm 1$ noise in §1-5 only to reduce notational clutter.

**Other concerns (Reviewer 4).** We acknowledge that our bounds in the proofs of Lemma 3.4 and Proposition 2 were sloppy but they can be fixed by only adding a further constant factor in the guarantees, that are absorbed in the big-oh. Apologies for the typo in the proof of Lemma 3.5. The values of $r$ and $h$ as reported in Section 7 are not mixed up (e.g. in Table 7). Despite the higher value of $r$ in this data set, the performance of the $ODTN_r$ algorithms is superior potentially as a result of the influence of the other dominating logarithmic factor.

**Improved presentation of §4 (Reviewer 2).** We agree that the presentation in §4 can include more details and better explanations. If the paper is accepted, we will use the extra content page for this.

[Meta-Review · NeurIPS 2019]

All reviewers are positive or very positive about the paper and most reviewers were satisfied by the authors reponse. This is a clear accept. I however encourage the authors to take into account the reviewers comments to improve their paper, especially the (unanswered) issues raised by reviewer 4.